# Staphylococcal toxin PVL ruptures model membranes under acidic conditions through interactions with cardiolipin and phosphatidic acid

Seong H. Chow[1,2], Yusun Jeon[1,2], Pankaj Deo[1,2], Amy T. Y. Yeung[3], Christine Hale[3,4], Sushmita Sridhar[3,4], Gilu Abraham[1,2], Joshua Nickson[1,2], François A. B. Olivier[1,2], Jhih-Hang Jiang[5,6], Yue Ding[1,7], Mei-Ling Han[5], Anton P. Le Brun[8], Dovile Anderson[9], Darren Creek[9], Janette Tong[1], Kip Gabriel[1], Jian Li[2,5], Ana Traven[1,2], Gordon Dougan[3,4], Hsin-Hui Shen[1,7]*, Thomas Naderer[1,2]*

1 Department of Biochemistry and Molecular Biology, Infection Program, Biomedicine Discovery Institute, Monash University, Clayton, Australia, 2 Centre to Impact AMR, Monash University, Clayton, Australia, 3 The Wellcome Sanger Institute, Wellcome Trust Genome Campus, Cambridge, United Kingdom, 4 Department of Medicine, Addenbrookes Hospital, Cambridge, United Kingdom, 5 Department of Microbiology, Infection Program, Biomedicine Discovery Institute, Monash University, Clayton, Australia, 6 Department of Infectious Diseases, Alfred Hospital and Central Clinical School, Monash University, Melbourne, Australia, 7 Department of Materials Science and Engineering, Faculty of Engineering, Monash University, Clayton, Australia, 8 Australian Centre for Neutron Scattering, Australian Nuclear Science and Technology Organisation, Kirrawee DC, Australia, 9 Drug Delivery Disposition and Dynamics, Monash Institute of Pharmaceutical Sciences, Monash University, Parkville, Australia

* hsin-hui.shen@monash.edu (H-HS); thomas.naderer@monash.edu (TN)

## Abstract

Panton-Valentine leukocidin (PVL) is a pore-forming toxin secreted by *Staphylococcus aureus* strains that cause severe infections. Bicomponent PVL kills phagocytes depending on cell surface receptors, such as complement 5a receptor 1 (C5aR1). How the PVL-receptor interaction enables assembly of the leukocidin complex, targeting of membranes, and insertion of a pore channel remains incompletely understood. Here, we demonstrate that PVL binds the anionic phospholipids, phosphatidic acid, and cardiolipin, under acidic conditions and targets lipid bilayers that mimic lysosomal and mitochondrial membranes, but not the plasma membrane. The PVL–lipid interaction was sufficient to enable leukocidin complex formation as determined by neutron reflectometry and the rupture of model membranes, independent of protein receptors. In phagocytes, PVL and its C5aR1 receptor were internalized depending on sphingomyelin and cholesterol, which were dispensable for the interaction of the toxin with the plasma membrane. Internalized PVL compromised the integrity of lysosomes and mitochondria before plasma membrane rupture. Preventing the acidification of organelles or the genetic loss of PVL impaired the escape of intracellular *S. aureus* from macrophages. Together, the findings advance our understanding of how an *S. aureus* toxin kills host cells and provide key insights into how leukocidins target membranes.

**Data availability statement:** All relevant data are within the paper and its Supporting Information files excluding raw flow cytometry files, which are freely accessible at http://flowrepository.org using the identifier "FR-FCM-Z9ZD".

**Funding:** The work was supported by the National Health and Medical Research Council (https://www.nhmrc.gov.au) Ideas Grant (1163556 to T. N. and H. -H. S) and Career Development Research Fellowship (GNT1106798 to H. -H. S.), an Australian Research Council (https://www.arc.gov.au) Discovery Early Career Researcher Award (DE140101788 to A.P.L.B., DE230100356 to M. -L. H.) and a Future Fellowship (FT170100313 to T. N.), and an Australian Institute of Nuclear Science and Engineering (https://www.ainse. edu.au) Post Graduate Research Award (to M.-L. H.). The funders had no role in study design, data collection and analysis, decision to publish, or preparation of the manuscript.

**Competing interests:** The authors have declared that no competing interests exist.

**Abbreviations:** ANSTO, Australian Nuclear Science and Technology Organization; bFGF, basic fibroblast growth factor; C5aR1, complement 5a receptor 1; CFU, colony-forming unit; CL, cardiolipin; CTG, Cell Tracker Green; DAG, diacylglycerol; EBs, embryoid bodies; gDNA, genomic DNA; HI, heart infusion; hiPSC-dMs, human-induced pluripotent stem cells-derived macrophages; HPLC, high-performance liquid chromatography; MβCD, methyl-beta-cyclodextrin; MCFC, myeloid-cell-forming complex; M-CSF, macrophage-colony-stimulating factor; MLV, multilamellar vesicles; MOI, multiplicity of infection; MRM, multiple reaction monitoring; NLRP3, NOD-like receptor protein 3; NR, neutron reflectometry; PA, phosphatidic acid; PBS, phosphate-buffered saline; PE, phosphatidylethanolamine; PC, phosphatidylcholine; PI, phosphatidylinositol; PMA, phorbol 12-myristate 13-acetate; PVL, Panton-Valentine leucocidin; PS, phosphatidylserine; QCM-D, Quartz-crystal microbalance with dissipative monitoring; SLC17A5, solute carrier family 17 member 5; SLD, scattering length density; SGMS1, sphingomyelin synthase 1; SM, sphingomyelin; TMRM, tetramethylrhodamine methyl ester; TSB, Trypticase Soy Broth; WT, wild type.

## Introduction

*Staphylococcus aureus* is a Gram-positive bacterium that causes a range of human diseases [1]. Skin and soft tissue infections as well as necrotizing pneumonia are associated with *S. aureus* strains expressing pore-forming toxins, such as Panton-Valentine leukocidin (PVL) [2]. PVL and other leukocidins primarily kill front-line phagocytes thus promoting bacterial survival during tissue invasion [3]. In addition, PVL triggers inflammation which contributes to the immunopathology of severe *S. aureus* infections [4]. PVL-expressing methicillin-resistant *S. aureus* strains, such as the USA300 clone, cause hospital outbreaks around the world and affect otherwise healthy individuals [5]. Despite the appreciation that leukocidins promote *S. aureus* infections, targeting the toxins has not improved outcomes in the clinic so far [6]. This may reflect redundancy between the different leukocidins but also our incomplete understanding how these toxins kill host cells.

PVL is secreted by *S. aureus* as two soluble monomers, termed LukS-PV and LukF-PV [7]. To kill host cells, PVL depends on the assembly of the subunits into an octameric complex that forms a membrane pore. The formation of a PVL pore complex is thought to compromise the integrity of the plasma membrane, leading to ion leakage and ultimately cell death [8]. Imaging the interactions between the subunits, and the formation of a membrane pore, however, has remained challenging, partly because PVL fails to rupture model membrane systems [9]. The structural details of PVL pore formation are largely inferred from related leukocidins, which spontaneously assemble into pore-forming complexes in solution [10,11]. More recently, protein receptors have been identified for each PVL subunit, whereby LukS-PV binds complement C5a receptor 1 (C5aR1) and LukF-PV the protein tyrosine phosphatase receptor C (also known as CD45) [12,13]. The interaction of PVL with the cell surface receptors explains the host and species specificity of the leukocidin, as both subunits bind human but not mouse receptors which are primarily expressed on phagocytes [3]. The expression of human C5aR1 enables PVL to bind phagocytes and induce cell death and inflammation in mice, although additional host factors are involved that contribute to the rapid cellular demise [12,14–16]. In particular, PVL induces host-programmed cell death factors, such as NOD-like receptor protein 3 (NLRP3), which activate host pore-forming proteins that rupture the plasma membrane [14,17]. How the interaction with host receptors enables the assembly of the PVL complex, toxin pore formation, and induction of cell death, however, remains largely unknown.

Here, we set out to characterize additional host factors that contribute to PVL activity in macrophages. By performing a genome-wide CRISPR-Cas9-based screen, we identified sphingomyelin biosynthesis as important host pathway for the cytotoxicity of PVL in human macrophages. Rather than enabling interaction with the plasma membrane, sphingomyelin, and cholesterol, promoted C5aR1 internalization after PVL interaction. Internalized PVL caused the disruption of lysosomes and mitochondria. PVL directly interacted with anionic phospholipids enriched in lysosomal and mitochondrial membranes. The PVL–lipid interaction enabled rupture of model membranes at acidic environments independent of proteinaceous receptors. Taken together, these findings advance our understanding how PVL interacts with host membranes and causes host cell death.

## Materials and methods

### Expression and purification of recombinant PVL and LukAB subunits

Polyhistidine-tagged LukAB, LukF-PV, and GST-tagged LukS-PV were expressed in *E. coli* strain BL21 star (DE3) harboring pETDuet1-LukAB, pET-18b-LukF, and pGEX-2T-LukS, respectively. The purified PVL subunits were cleaved by thrombin at 25°C for 24 h, followed by thrombin removal using *p*-aminobenzamidine agarose (Sigma-Aldrich). The endotoxin in purified PVL subunits and LukAB was removed to a final concentration of <0.1 EU/mg of purified protein using High Capacity Endotoxin Removal Resin (Pierce).

### Bacterial strains and growth

*S. aureus* strains SF8300 (USA300) are methicillin-resistant clinical isolates from the United States. The wild type SF8300 strain harboring DsRed or cGFP constitutively expression vector was generated using the plasmid pHC48 [18]. The Δ*pvl* deletion strain and the complemented Δ*pvl* strain were generated from *S. aureus* SF8300 strain and were kindly donated by A/Prof Binh Diep (UCSF, School of Medicine) [19].

   *S. aureus* strains were grown in heart infusion (HI) broth at 37°C overnight, and then grown in fresh CCY media (3% [w/v] yeast extract, 2% [w/v] Bacto-Casaminoacids, 2.3% [w/v] sodium pyruvate, 0.63% [w/v] $Na_2HPO_4$, and 0.041% [w/v] $KH_2PO_4$, pH6.7) at 37°C to reach mid-log phase. The bacterial numbers before infection were determined in 1 ml phosphate-buffered saline (PBS) at $OD_{600nm}$, where an $OD_{600nm}$ of 1 corresponds to $1 \times 10^9$ bacteria per ml, which was used to determine the multiplicity of infection (MOI).

### Generation of cGFP- and DsRed-expressing *Staphylococcus aureus* for macrophages infection assay

To prepare *S. aureus* SF8300 (USA300) strain competent cells, the bacteria was grown in Trypticase Soy Broth (TSB) (Oxoid) at 37°C overnight. The overnight culture was diluted into 100 ml TSB to $OD_{578 nm} = 0.5$, and incubated at 37°C, 200 rpm for 40 min to reach $OD_{578 nm} = 0.7$. The cultures were then chilled in ice, and then harvested by centrifugation at 4,000 × g for 10 min. The cell pellet was resuspended in 100 ml ice-cold water, and centrifuged at 4,000 × g for 10 min. The cell pellet was then resuspended in 10 ml 10% (v/v) glycerol, and centrifuged at 4,000 × g for 10 min. Then, the cell pellet was resuspended in 4 ml 10% (v/v) glycerol, and centrifuged again at 4,000 × g for 10 min. The cell pellet was resuspended in 250 μL 10% (v/v) glycerol, aliquoted into 50 μL per vials, and stored at −80°C until required.

   The competent cells were thawed at room temperature, and centrifuged at 5,000 × g for 1 min. The cell pellet was resuspended in 80 μL 10% (v/v) glycerol, 500 mM sucrose in MiliQ water. 10 μL plasmid (pHC48, 200 ng/μL) was added to the cells, and electroporated at 2.5 kV, 100 Ω, 25 μF. Then, 1 ml HI broth supplemented with 500 mM sucrose was immediately added to the cells, and incubated at 37°C, 200 rpm for 1 h. The cells were plated on HI agar supplemented with 10 μg/ml chloramphenicol, and incubated at 37°C for 24–36 h.

   *S. aureus* (SF8300) harboring the pHC48 construct was grown in HI broth supplemented with 10 μg/ml chloramphenicol at 37°C overnight. The culture was diluted in 1:100 and grown in fresh CCY media at 37°C to reach mid-log phase. The bacterial numbers before infection were determined in 1 ml PBS at $OD_{600 nm}$, where an $OD_{600 nm}$ of 1 corresponds to $1 \times 10^9$ bacteria per ml, which was used to determine the MOI.

### THP-1-differentiated human macrophages

THP-1 human monocytes were cultured in RPMI 1,640 media supplemented with 10% FBS and 25 mM HEPES (Sigma). Differentiation into macrophages was induced with 80 nM of phorbol 12-myristate 13-acetate (PMA) (Sigma) in tissue culture-treated multi-well plates at a density of $5 \times 10^5$ cells/ml over 24 h. Adherent macrophages were subsequently cultured in fresh media without PMA for an additional 24 h before the experiments.

## Human-induced pluripotent stem cells-derived macrophages (hiPSC-dMs)

Macrophage differentiation was initiated from hiPSCs as described recently [20,21]. Briefly, hiPSCs were cultured in Vitronectin XF-coated 6-well plate with complete TeSR-E8 media (StemCell Technologies). Once compact, round-shaped colonies formed, the colonies were harvested with Gentle Dissociation Buffer (StemCell Technologies) and co-cultured with mouse feeder cells (gamma-irradiated MEF cells) in advanced DMEM/F12 media (Gibco) supplemented with 20% knockout serum, glutaMAX, 55 mM β-mercaptoethanol and 4 ng/ml basic fibroblast growth factor (bFGF) (R&D System). The hiPSCs were then harvested and cultured in the advanced DMEM/F12 media without bFGF, to form spherical embryoid bodies (EBs) after 4 days. Next, the EBs were differentiated in X-VIVO 15 media (Lonza) supplemented with glutamax, 55 mM β-mercaptoethanol, 25 ng/ml Interleukin-3 (IL-3), and 50 ng/ml macrophage-colony-stimulating factor (M-CSF) to form endothelial-like stromal cells and myeloid-cell-forming complex (MCFC) over 21 days. Suspension cells released from MCFCs were further differentiated into macrophages-like cells in RPMI media supplemented with 10% FBS, glutamax, 55 mM β-mercaptoethanol, 25 mM HEPES, and 100 ng/ml M-CSF for 7 days.

## CRISPR/Cas9-mediated genome-wide screen

The CRISPR/Cas9 genomic screen of host genes was adapted from A. T. Y. Yeung and colleagues [22]. A total of $2.4 \times 10^7$ Cas9-THP-1 cells were transduced with the GeCKO v2.0 half library A at an MOI of 0.3 and selected in puromycin for 14 days. Subsequently, the cells were differentiated to macrophages by exposure to PMA and treated with PVL (2 µg/ml). Viable macrophages were collected by cell sorting and genomic DNA (gDNA) was extracted. PCR was performed to amplify the gRNA regions, and subsequently sequencing adaptors and barcodes were attached to the samples. Samples were sequenced on an Illumina HiSeq2500 for 50-bp single-end sequencing. The numbers of reads for each guide were counted with an in-house script. Enrichment of guides and genes was analyzed using MAGeCK statistical package version 0.5.2, as described previously [20].

## Generation of CRISPR/Cas9 mutant THP-1 macrophages

Candidate genes selected from the mutant library screen results were validated by generating targeted mutants in Cas9-THP-1 cells using sgRNAs. Four sgRNAs per candidate gene were designed as described previously [20]. The sgRNAs (IDT) were cloned into the lentiviral gRNA expression vector, pKLV-U6gRNA(BbsI)-PGKpuro2ABFP [23]. HEK293T cells were co-transfected with the pLenti constructs and ViraPower packing mix according to the manufacturer's protocol (Invitrogen). The medium was replaced with fresh medium 24 h after transfection. Viral supernatant was harvested 48 h after transfection and stored at −80°C. Cas9-THP-1 cells were transduced with the viral supernatants as described above and selected in 0.9 µg/ml puromycin (Sigma) for 2 weeks. Lentivirus expressing the empty lentivirus vector was also transduced into Cas9-THP-1 cells as control. The resultant mixed population of mutants generated from each gRNA was screened for resistance to PVL using live-cell imaging. Subsequently, 2 gRNAs per candidate gene were chosen, and samples were single-cell sorted into 96-well plates using FACS Aria II Fluorescence-activated cell sorter to achieve 96 clonal mutant lines per gRNA. Clonal mutant lines were further expanded and screened by live-cell imaging for resistance to PVL. To confirm the *sgms1* gene mutation, the genomic region harboring the gRNA cutting site was amplified using primers in S1 Table and was submitted for sequencing.

## Lipid analysis

THP-1 macrophages ($2 \times 10^7$ cells) were washed with ice-cold PBS and the cell pellet was snap frozen. The cell pellets were resuspended in 1 ml chilled PBS, then centrifuged at 2,000 g for 5 min at 4°C, and the cell pellet resuspended in 200 µL of extraction buffer (chloroform:methanol:water, 2:6:1 (v/v)). The samples are shaken vigorously at 4°C for 1 h and centrifuged at 14,800 g for 5 min at 4°C. The supernatant (160 µL) was transferred to new tubes and the extraction solvent

was evaporated under the stream of nitrogen at 20°C (Biotage evaporator). On the day of analysis, the samples were dissolved in 40 μL of butanol:methanol:water (4.5: 4.5: 1, v/v) mixture. The samples were sonicated in a water bath for 30 min at the temperature below 30°C, and vortexed on a rotary vortex for 15 min. The samples were centrifuged at 14,800 g for 10 min at 20°C and transferred to LC-MS vials. LC-MS analysis was performed on a Q-Exactive Orbitrap mass spectrometer (Thermo Fisher Scientific) coupled with high-performance liquid chromatography (HPLC) system Dionex Ultimate 3000 RS (Thermo Fisher Scientific). 10μl were injected and the column was run for 30 min. Samples were analyzed in the full scan mode with positive and negative polarity switching at 70,000 resolution at 200 m/z with detection range of 140–1,300 m/z. Lipid species were identified based on mass and retention time using internal standards. Quantification was performed using multiple reaction monitoring (MRM) and semi-automated peak integration using Tracefinder 3.2 (Thermor Fisher) with manual verification.

## Monitoring lysosomal and mitochondrial integrity and macrophages cell death using live-cell imaging

Cells were plated in tissue culture plates at $5.0 \times 10^4$ cells/well (96 wells) overnight. For monitoring lysosomal integrity, cells were stained with 1 μM Cell Tracker Green (CTG, ThermoFisher Scientific) for 30 min in serum-free RPMI 1640. Cells were then replaced with culture medium containing 600 nM Draq7 (Abcam) and 50 nM LysoTracker Red (ThermoFisher Scientific) with or without inhibitors Ca-074Me or Bafilomycin A1 (Sigma-Aldrich) for 30 min. For monitoring mitochondrial integrity and macrophages cell death, cells were stained with 1 μM Cell Tracker Green (CTG, ThermoFisher Scientific) for 30 min in serum-free RPMI 1640. Cells were then replaced with culture medium containing 600 nM Draq7 (Abcam) and tetramethylrhodamine methyl ester (TMRM, ThermoFisher Scientific) with or without inhibitors Ca-074Me or Bafilomycin A1 (Sigma-Aldrich) for 30 min. Culture medium containing purified PVL (62 ng/ml or as indicated otherwise), LukS-PV (31 ng/ml), LukF-PV (31 ng/ml), LukAB (15.6 ng/ml), or nigericin (Sigma) was added to the cells and immediately imaged.

Live-cell imaging was performed on a Leica DMi8 widefield fluorescent microscope equipped with an incubator chamber set at 37°C, 5% $CO_2$, and an inverted, fully-motorized stage driven by Leica Advanced Suite Application software. Time-lapse images were acquired with bright-field, GFP and Y5 filters every 30 min for up to 5 h using a 10×/0.32-NA objective lens. To determine the percentage of dead cells, images were analyzed in ImageJ and in MetaMorph (Molecular Devices) using a custom-made journal suite incorporating the count nuclei function to segment and count the number of CTG, Lysotracker Red, TMRM and Draq7-positive cells as described previously [24]. CTG staining was used to calculate the total cell number in each sample. The percentage of Lysotracker Red, TMRM, and Draq7-positive cells was analyzed in Excel and GraphPad Prism 9.0.

## In vitro *Staphylococcus* infection assays

To determine bacterial burdens, PMA-differentiated macrophages were seeded at a density of $1.0 \times 10^6$ cells per ml into 24-well tissue culture plates. Prior to infection, the macrophages were pretreated with DMSO or Bafilomycin A1 (100 nM) for 4 h, and infected with *S. aureus* strains (wild type, Δ*pvl* deletion strain and the complemented Δ*pvl* strain, or DsRed expressing strain) at a MOI of 10. After 30 min, macrophages were washed three times in PBS, incubated in RPMI complete media supplemented with 10 μg/ml Lysostaphin (Sigma-Aldrich) for 40 min. The macrophages were washed three times in PBS, and then treated with DMSO or Bafilomycin A1 (100 nM), as indicated. Then, culture supernatant was collected and cells were lysed in 1% (w/v) Saponin in PBS for 5 min at room temperature. Serial dilutions of the cell lysates and supernatant were plated on HI agar plates, and bacterial colonies were counted after 16 h at 37°C.

## Monitoring in vitro *Staphylococcus* infection using live-cell imaging

Cells were plated in tissue culture plates at $1.0 \times 10^5$ cells/well (96 wells) overnight. Prior to infection, cells were stained with 1 μM Cell Tracker Green (CTG, ThermoFisher Scientific) for 30 min in serum-free RPMI 1640. The cells were then

pretreated with DMSO or Bafilomycin A1 (100 nM) for 4 h, and infected with *S. aureus* strains (wild type, Δ*pvl* deletion strain, and the complemented Δ*pvl* strain, or DsRed expressing strain) at a MOI of 10. After 30 min, macrophages were washed three times in PBS, incubated in RPMI complete media supplemented with 10 µg/ml lysostaphin (Sigma-Aldrich) for 40 min. The macrophages were washed three times in PBS, and then treated with culture medium containing 50 nM TMRM and 600 nM Draq7 (Abcam) with or without Bafilomycin A1, and immediately imaged. For monitoring DsRed-expressing *Staphylococcus* strain, cells were stained with 600 nM Draq7 (Abcam) with or without Bafilomycin A1, and immediately imaged. Live-cell imaging was performed on a Leica DMi8 widefield fluorescent microscope equipped with an incubator chamber set at 37°C, 5% $CO_2$, and an inverted, fully-motorized stage driven by Leica Advanced Suite Application software. Time-lapse images were acquired with bright-field, GFP and Y5 filters every 30 min for up to 12 h using a 10×/0.32-NA objective lens. The images were analyzed in ImageJ.

## Immunoblotting

Cell lysate (reduced and denatured) was separated on 15% SDS-PAGE gels, and proteins were wet-transferred to nitrocellulose membranes (BioRad) for detection. Membranes were blocked with TBS containing 0.2% Tween 20 (TBS-T) containing 5% skim milk for 1 h and were then probed with primary antibodies to LukS-PV (made inhouse), human C5aR1 (clone B-6, #sc-271949, Santa Cruz Biotechnology) at 4°C overnight. The membranes were probed with relevant HRP-conjugated secondary antibodies for 1 h. Membranes were washed three times in TBS-T between antibody incubations and all antibodies were diluted in TBS-T containing 5% skim milk. Membranes were developed using ECL (Thermo Fisher Scientific) and exposed to film (Kodak).

## Immunofluorescence staining assay and confocal imaging

Macrophages were seeded onto glass coverslips in 24-well plates and treated with specific inhibitor for 30 min prior to PVL treatment for indicated time. For plasma membrane staining, cells were stained with Cytopainter Deep Red (#ab219942, Abcam) for 5 min at 37°C prior to PVL treatment for indicated time. Cells were then fixed with 4% (w/v) PFA for 20 min at room temperature, washed trice with PBS, and treated with 50 mM $NH_4Cl$ for 10 min. Cells were permeabilised in 0.1% (v/v) Triton-X 100 in PBS for 10 min, washed trice with PBS, and incubated with blocking buffer (PBS with 3% w/v BSA) for 1 h at room temperature. Cells were then incubated with primary antibody to LukS-PV (made inhouse), human C5aR1 (clone B-6, #sc-271949, Santa Cruz Biotechnology), human LAMP1 (clone H4A3, DSHB), or anti-gal3-A647 (Thermo Fisher, cat#125,408) for 30 min at room temperature. After three washes in PBS, cells were incubated with blocking buffer containing Alexa Fluor conjugated goat secondary antibodies (Life Technologies) if necessary and 0.1 µg/ml DAPI (Sigma) for 30 min. After three washes in PBS, coverslips were mounted on glass slides with Fluorescence Mounting Medium (Dako) for overnight at room temperature and sealed with nail polisher. The slides were imaged on Leica SP5 confocal microscope using a 40×/1.25 NA objective lens. Images were analyzed in ImageJ [25].

For super-resolution confocal imaging (S12B Fig), THP-1 cells were seeded in Starstedt 8-well glass chamber slides at $1 \times 10^5$ cells/ml density, 0.5 ml medium per well (Starstedt, #94.6190.802). Eighty nM of PMA was added, and cells were incubated at 37°C, 5% $CO_2$ overnight. Cells were further incubated without PMA for 24 h. Cells were treated with 100 nM of BafA1 (or DMSO control vehicle) for 1 h, prior to exposure to PVL (LukF and PVL LukS at 62.5 ng/ml each) or 500 µM LLO-Me. After 1 h, chamber wells were rinsed with PBS three times, and cells fixed with 4% (w/v) PFA for 20 min at room temperature. Cells were then permeabilised with 0.3% (v/v) Triton X-100 in PBS and incubated at room temperature for 10 min. Blocking buffer was then added: 3% (w/v) BSA in PBS with 0.1% (v/v) Triton X-100, and cells were incubated at room temperature for 1 h. Cells were then incubated with anti-gal3-A647 (Thermo Fisher, #125408) in PBS with 0.1% (v/v) Triton X-100 and 1% (w/v) BSA for 30 min at room temperature. From this step onwards, the glass chamber slides were covered with foil in between cell treatment steps. Cells were rinsed with PBS three times, then stained with DAPI (0.1 µg/ml in PBS with 0.1% v/v Triton X-100 and 1% w/v BSA) and incubated for 30 min at room temperature. Wells were then

rinsed with PBS three times before confocal imaging using a Zeiss LSM 980 Airyscan system with a 37°C cage incubator. 8-bit, 2245 × 2245 px frame size, 1.7× crop area images were captured using an 63×/1.4 NA objective lens and an Airyscan 2 detector. Images were processed using Image J 1.54f [25]. Gal3 images (red) have been displayed with a 65% decrease in intensity maximum value (all images), and DAPI images (blue) have had a 15% increase in intensity minimum value (all images).

## Protein-lipid overlay assay

Membrane lipid strips (Echelon) were incubated with blocking buffer (TBS-T containing 3% w/v BSA) for 1 h at room temperature. Membranes were then incubated with purified proteins (10 μg/ml) for 1 h at room temperature, followed by washing trice with TBS-T for 5 min. Next, membranes were probed with primary antibody against LukS-PV (made inhouse) for 1 h at room temperature and were washed trice with TBS-T for 5 min. After washing, membranes were probed with secondary goat anti-rabbit IgG (Sigma) antibodies conjugated HRP in blocking buffer. Membranes were developed with the ECL reagent (Clarity ECL, BioRad) before acquiring image with ChemiDoc Touch Imaging System (BioRad). Acquired images were then exported using Image Lab (BioRad) and processed using Affinity Designer (Serif).

## Unilamellar vesicles

The protocol was adapted from Y. Ding and colleagues [26]. Briefly, 1-palmitoyl-2-oleoyl-glycero-3-phosphocholine (POPC, Avanti), POPC/1-palmitoyl-2-oleoyl-snglycero-3-phospho-L-serine (POPS) (POPC/POPS, *w:w* 2:1), POPC/1-palmitoyl-2-oleoyl-sn-glycero-3-phosphate (POPA) (POPC/POPA, *w:w* 2:1), POPC/18:1 cardiolipin (TOCL) (POPC/TOCL, *w:w* 4:1), and POPC/Sphingomyelin(SM)/Cholesterol (Cho) (POPC/SM/Cho, *w:w* 4:1:1) were dissolved in chloroform in round-bottom glass test tubes and then were dried under a gentle stream of pure nitrogen gas with continuous rotation. The test tubes were then freeze-dried overnight to remove residual chloroform. Lipid films were rehydrated and vortex mixed vigorously in buffer solution containing 10 mM HEPES, pH 7.4, and 150 mM NaCl to form large multilamellar vesicles (MLV) at a final lipid concentration of 1 mg/ml. Unilamellar vesicles (LUV) were formed by extrusion of the multilamellar vesicles using an Avanti Mini-extruder (Avanti Polar Lipids) with polycarbonate membranes of 100 nm pore size (Whatman). The cloudy MLV solution was passed through the extruder until clear.

## Liposome leakage assay

The lipid films (1 mg) were hydrated with 1 ml hydration buffer containing sulforhodamine B solution (10 mM HEPES, pH 7.4, 150 mM NaCl, 50 mM sulforhodamine B) for 1 h above the $T_m$ of the lipids. The liposomes were vortexed regularly while hydrating, and then sonicated in bath sonication for 5–10 min above the Tc of the lipids. The liposomes were washed twice to remove the external dye with 1 ml hydration buffer (10 mM HEPES, pH 7.4, 150 mM NaCl) by centrifugation in a TLA-120.2 rotor (Beckman) at 4°C for 30 min at 55,000 rpm. After centrifugation, the liposomes were resuspended in either 1 ml neutral pH buffer (10 mM HEPES, pH 7.4, 150 mM NaCl) or acidic pH buffer (0.1 M citric acid, 0.2 M $Na_2HPO_4$, pH 5.0). For the time course liposome leakage assay, aliquots of 300 μM sulforhodamine B encapsulated liposomes were mixed with LukF-PV, LukS-PV (500 ng/ml), or PVL (1 μg/ml) recombinant proteins in 100 μL neutral or acidic pH buffer followed by 30 min incubation at room temperature. The excitation and emission wavelengths were 565 and 586 nm, respectively. The fluorescence emission was continuously recorded for 30 min at 30 s intervals, then 0.1% (v/v) Triton X-100 was added to achieve complete release.

## Quartz-crystal microbalance with dissipative monitoring (QCM-D)

The protocol was adapted from J. Lu and colleagues [27] using the Q-Sense E4 QCM-D system. The QCM-D instrument was equipped with an axial flow chamber, IPC High Precision Peristaltic Multi-Channel Dispenser (ISMATEC, flow rate 0.2 ml/min), and held at a constant temperature of 25°C during the experiments. The silicon oxide-coated sensor crystals

(SiO$_2$, 50 nm thickness, Q-Sense) were cleaned by immersion of the surfaces in 10 mM SDS for 8 h, rinsing extensively with Milli-Q water, drying with a gentle stream of pure nitrogen gas, and oxidation in a UV-ozone chamber for 15 min to remove residual organic impurities immediately before use. On adsorption of lipid bilayers consisting of POPC/POPA (w/w 2:1) or POPC/TOCL (w/w 2:1) and purified PVL (100 µg/ml) in different pH buffers by the sensor surface, shifts in resonance frequency (*f*) and changes in energy dissipation (*D*) were obtained and analyzed with QSoft401 V2.5.2.418 software. *f* reflects the mass of adsorbed film (including coupled water) on the sensor surface and *D* correlates with the viscoelastic properties of molecular layers on the surface of the sensor [28]. *f* and *D* were monitored simultaneously at the fundamental natural frequencies 15, 25, and 35 MHz, which corresponding to third, fifth, and seventh harmonics (overtones) of the 5 MHz crystal. The frequency change (Δ*f*) of oscillating quartz could be linearly related to its mass change (Δ*m*) as expressed by Sauerbrey equation: $\Delta m = -C \times 1/n \times \Delta f$, where n is the overtone number and C is a constant, which is approximately equal to 17.7 ng/(cm$^2$·Hz) for a 5 MHz AT-cut quartz crystal at room temperature.

## Neutron reflectometry (NR) measurement

NR data were collected using the Platypus time-of-flight neutron reflectometer that used a cold neutron spectrum (2.8 Å ≤ λ ≤ 18.0 Å) from the OPAL 20MW research reactor (Australian Nuclear Science and Technology Organization (ANSTO), Sydney, NSW, Australia) [29]. A chopper pairing of choppers 1 and 4 set to a 24 Hz rotation speed was used and provides a wavelength resolution (Δλ/λ) ~8%. Using a vertical scattering geometry, the neutron beams reflected from the sample interface were collected at two glancing angles of incidence (0.85° for 300 s and 3.5° for 3,600 s) to cover the momentum transfer (Q) range of interest (0.01 Å$^{-1}$ ≤ Q ≤ 0.3 Å$^{-1}$), where Q is defined as: $Q = 4\pi \sin(\theta)/\lambda$, where θ is the angle of incidence and λ the wavelength. An illuminated footprint of 33 mm wide and 50 mm long was used. Silicon wafers used were 4-inch wafers 10 mm thick (orientation 110, p-type) and polished on one side. The polished side of sample silicon wafer faced a roughened silicon wafer which had in-let and out-let holes for solvent exchange and the two wafers were separated by a 100-µm-thick PTFE gasket. The sample and backing wafers were held together using aluminum plates that were bolted together.

Initially, 10 ml pD 7.4 buffer solution (10 mM HEPES, 150 mM NaCl, 5 mM CaCl$_2$) containing 30 µg LUV was added to the silicon sample wafer. The supported membrane lipid bilayer was characterized in pH/D 7.4 buffer solution with three isotopic contrasts with scattering length density (SLD, ρ), D$_2$O (99.9%, $\rho_\_ = 6.35 \times 10^{-6}$ Å$^{-2}$), CmSi (contrast-matched silicon, 38% D$_2$O: 62% H$_2$O v/v; $\rho_\_ = 2.07 \times 10^{-6}$ Å$^{-2}$), and H$_2$O ($\rho_\_ = -0.56 \times 10^{-6}$ Å$^{-2}$) to confirm the successful formation of the model membrane. For the exchange between these three contrasts, a total of 10 ml of pH/D 7.4 buffer solution (10 mM HEPES, 150 mM NaCl, 5 mM CaCl$_2$) was pumped through the sample wafer at the rate of 1.0 ml/min using an HPLC pump (Knauer GmbH, Berlin, Germany). Then, either 3 ml neutral pD buffer (10 mM HEPES, pD 7.4, 150 mM NaCl) or acidic pD buffer (0.1 M citric acid, 0.2 M Na$_2$HPO$_4$, pD 5.0) containing purified PVL (100 µg/ml) was injected to the sample wafer and incubated for 1 h, which was subsequently washed out by the injection of D$_2$O buffer to remove any unspecific bindings using an HPLC pump. The PVL membrane was then fully characterized under the three isotopic contrasts.

Final reduced data were obtained by combining the data from the two angles together using in-house reduction software, SLIM [30], that accounts for detector efficiency and converts the time-of-flight data to wavelength. The software is used to calculate Q, re-bin the data to instrument resolution, stitch the datasets from the two angles of incidence at the overlap region to provide a complete reflectivity profile and scale the data to adjust the critical edge equal to a reflectivity of one.

## NR data analysis

The MOTOFIT analysis program [31] in IGOR PRO 7.08 software (WaveMetrics , Portland, OR, USA) was used to analyze the NR profiles. The data are described as a series of layers with each sublayers fitted according to thickness, SLD, and roughness using optical matrix method [32]. The thickness and SLD were varied using least-squares regression with a genetic algorithm until the best fit matched the data with an acceptable χ$^2$ value. The reflectometry data of three isotopic

contrasts ($D_2O$, CmSi, and $H_2O$) were then fitted simultaneously (co-refinement) with the thickness and roughness constrained to be the same across each contrast and only the SLD allowed to vary. The SLD is defined as the sum of the scattering lengths for each component ($b_i$) divided by the molecular volume ($V_m$):

$$SLD = \frac{\sum_{i=1}^{n} b_i}{V_m}$$

and is analogous to a neutron refractive index. Error analysis was conducted by running Monte Carlo routines using the best fit which generate 1,000 fits of the data within the limits of each parameter. Each fitted parameter is then histogrammed and the median and standard deviation of the distribution calculated. From the fitted data the volume fraction of each component in the sublayers was shown as a percentage and was calculated by comparing the fitted SLD against the theoretical SLD (S2 Table). When a layer is composed of two components, namely a chemical species $s$ and water $w$, the resultant SLD ($\rho$) can be given by: $\rho layer = \varphi \rho_s + (1 - \varphi)\rho_w$, where $\rho_s$ and $\rho_w$ are the theoretical SLD of the two components (S2 Table), respectively, and $\varphi$ is the volume fraction of chemical species $s$ in the layer. The volume fractions of lipid bilayer were determined by the volume fractions of the lipid tails region of the bilayer. The coverage of PVL in their binding layers was determined by comparing the fitted SLD values against the calculated SLD values of PVL, lipids, and water.

### Statistical analysis

For two-group analysis, an unpaired, parametric $t$ test was applied. Multiple-group comparisons analysis was based on a one-way ANOVA. All tests were analyzed in GraphPad Prism 9.0, and $P$ values less than 0.05 were considered statistically significant. The number of biological replicates and the statistical values for the individual experiments are stated in the figure legends.

### Results

#### PVL binds to lipid bilayers containing anionic phospholipids at acidic pH

While host lipids can promote the activity of bacterial toxins, such as cholesterol-dependent cytolysins, little is known whether PVL interacts with specific lipids and whether this contributes to membrane rupture. To ascertain whether PVL binds to any of the major lipid classes of mammalian cells, we initially performed a protein-lipid overlay assay. We did not detect binding of PVL to lipids enriched in the plasma membrane, such as phosphatidylcholine, sphingomyelin, or cholesterol (Fig 1A). PVL, and to some degree LukS-PV by itself, however, interacted with phosphatidic acid (PA), phosphatidylserine (PS), and cardiolipin (CL), which are the major anionic phospholipids in mammalian cells (Fig 1A). To verify the interaction of PVL and anionic phospholipids, we next generated model membranes containing ratios of phosphatidylcholine (PC) and PA, cardiolipin, or PS to mimic mammalian membranes. Surprisingly, PC, PC/PS, PC/PA, and PC/CL containing membranes failed to bind PVL as detected in real time using quartz-crystal microbalance with dissipative monitoring (QCM-D) (Fig 1B and 1C, S1A and S1B Fig, indicated with arrow 3). PA contributes only a small fraction of the total cellular lipid content but is an important signaling molecule and enables membrane interactions of effector proteins including in lysosomes [33]. We thus speculated that PVL–lipid interactions may dependent on the environmental pH. Consistent with this notion, reducing the pH to 5.0 of the incubation media to mimic the acidic environment of lysosomes promoted PVL binding to PC/PA, but not PC, containing membranes (Fig 1B and S1A Fig, arrow 3), as a decrease in the frequency ($f$) corresponds to increased mass uptake at the surface and a dissipation ($D$) correlates with viscoelasticity of the surface. PVL failed to markedly bind to PC/PS membranes under acidic conditions (S1B Fig, arrow 3). Similar to PA, however, PVL interacted with CL containing membranes at acidic but not neutral pH (Fig 1C). Cardiolipin is enriched in the inner membrane of mitochondria. To test whether PVL would gain access to the inner mitochondrial membrane, LukS-PV was incubated with isolated mitochondria (S1C Fig). Notably, purified recombinant LukS-PV interacted with mitochondria,

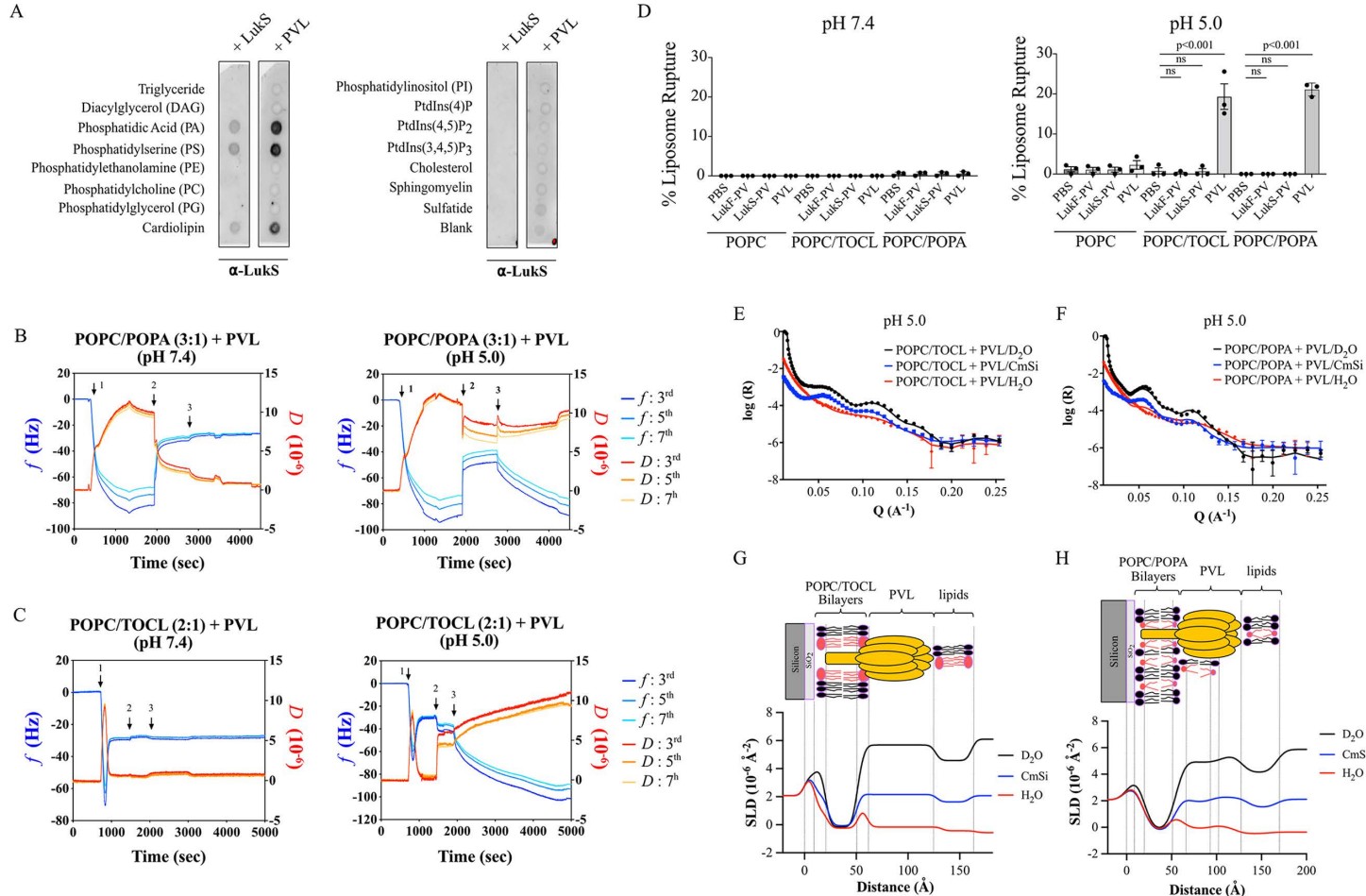

**Fig 1. PVL forms a protein complex on supported lipid bilayers containing anionic phospholipids at acidic pH. (A)** Protein-lipid overlay assay using purified LukS-PV or PVL (10 μg/ml; equal molar concentration of LukS-PV and LukF-PV) and probed with anti-LukS-PV antibodies. Data representative of two independent experiments. **(B–C)** Quartz-crystal microbalance with dissipative monitoring (QCM-D) assays using lipid bilayer of **(B)** POPC/POPA (w/w 2:1) or **(C)** POPC/TOCL (w/w 2:1), added at indicated 1↓, exchanged buffer to neutral or acid pH at 2↓ and exposed to PVL (100 μg/ml) at 3↓. Blue: frequency; red: dissipation. Data representative of two independent experiments. **(D)** Sulforhodamine B encapsulated liposomes of POPC, POPC/TOCL (w/w 2:1), or POPC/POPA (w/w 3:1) were treated with purified PVL (1 μg/ml) or its subunits (LukS-PV, LukF-PV) at neutral or acidic pH and fluorescence was determined after 30 min, compared to control Triton X-100 treatment. Mean ± SEM from three independent experiments, *P* values (one-way ANOVA followed by Dunnett's multiple Comparison test) are shown. **(E–F)** Neutron reflectometry (NR) measurements of purified PVL (100 μg/ml) exposed to **(E)** POPC/TOCL (w/w 2:1) and **(F)** POPC/POPA (w/w 2:1) membrane in pH 5.0 buffer. Black curve: measurement in $D_2O$ buffer; Blue curve: measurement in contrast matched Si buffer (CmSi); Red curve: measurement in $H_2O$ buffer. **(G–H)** Scattering length density profiles with cartoons for PVL with **(G)** POPC/TOCL and **(H)** POPC/POPA membranes. Black line: measurement in $D_2O$ buffer; Blue line: measurement in contrast matched Si buffer (CmSi); Red line: measurement in $H_2O$ buffer. The data underlying this Figure can be found in S1 Data.

and importantly was protected from proteinase treatment, similar to the inner membrane protein, Tim23 (S1C Fig). In contrast, the outer membrane protein Tom70 was cleaved after proteinase K treatment (S1C Fig). This demonstrates that LukS-PV is imported into isolated mitochondria, enabling interaction with cardiolipin. Taken together, these results suggest that PVL interacts with membranes containing phosphatidic acid and cardiolipin under acidic conditions.

## PVL complex formation on lipid bilayers containing anionic phospholipids at acidic pH

To functionally investigate whether PVL depends on host lipids to form pores, we next determined toxin activity using a liposome leakage assay. As expected, PC-containing liposomes remained intact after PVL treatment at both acidic and

neutral pH (Fig 1D and S2A Fig). Notably, PC/CL or PC/PA-containing liposomes ruptured after PVL treatment at pH 5.0, but not at pH 7.4 (Fig 1D, S2B and S2C Fig). The leukocidin subunits alone failed to cause liposome leakage (Fig 1D, S2A, S2B, and S2C Fig). This suggests that the two soluble subunits of PVL interact to rupture model membranes depending on the lipid and environmental conditions.

Given our observation that PVL can bind and rupture membranes independent of proteinaceous receptors, we next determined whether PVL was able to from a protein complex on model membranes. The PVL–lipid interaction was imaged using NR which enables molecular resolution of protein and lipid molecules on solid supports such as silicon surfaces. The CL- and PA-containing membranes were incubated with three isotopic solvents ($H_2O$, $D_2O$, and CmSi), and the reflectivity profiles were modeled simultaneously to determine the bilayer structures (S3 Fig). The CL-containing bilayer resulted in a single broad fringe at pH 7.4 and 5.0 (S3A, S3B, S3E, and S3F Fig), whereas an excess of additional lipid material was observed above the PA-containing bilayer even after excessive washing (S3C, S3D, S3G, and S3H Fig). The reflectivity profile in three different isotopic solvents ($H_2O$, $D_2O$, and CmSi) demonstrated protein-lipid interactions when PVL was added to CL- and PA-containing membranes at pH 5.0, but less so at neutral pH (Fig 1E and 1F, S4 and S5 Fig). The binding of PVL to CL- and PA-containing membranes at acidic pH resulted in distinct multiple fringes that corresponded to layers of lipids and/or proteins (S4B and S4D Fig). The distal CL-containing membrane head group layer contained volume fractions of 50.9 ± 7.9% phospholipid and 17.7 ± 2.6% PVL, besides water (Fig 1E and 1G and S3A Table). Similar coverage of PVL was observed immediately adjacent to the membrane, where the thickness of PVL was 70.9 ± 2.0 Å (Fig 1G and S3A Table), which is in agreement with the crystal structure of the γ-hemolysin complex [34] and the neutron structure of the α-hemolysin complex [35]. We also detected PVL within the phospholipid fatty acyl tail layer at a coverage of 2.5 ± 1.1% (S3A Table). Additional lipids were detected posterior to PVL corresponding to 22.9 ± 3% coverage, a similar value as for the PVL coverage (Fig 1G and S3A Table). At neural pH, there was only 2.0 ± 0.5% coverage of PVL above the CL-containing lipid bilayers with no PVL contribution to the phospholipid fatty acyl tail layer (Fig 1H and S3B Table). In addition, the thickness of PVL was 34.4 ± 2.3 Å adjacent to the lipid bilayer at pH 7.4 (S5C Fig and S3B Table), suggesting lack of complex formation. Similar results were obtained using PA-containing membranes, whereby acidic pH increased protein-lipid interactions compared to neutral pH (Fig 1F and 1H, S4C and S4D Fig and S4A Table). As observed with cardiolipin-containing membranes, the predicted thickness of PVL was 68.1 ± 7.2 Å and covered 25.9 ± 1.1% of the layer (Fig 1H and S4A Table). PVL was also detected within the phospholipid tail layer of the membrane with a volume fraction of 2.1 ± 0.9% (Fig 1H and S4A Table). In contrast to the 25% coverage at acidic pH, PVL accounted for not more than 4% in each lipid layer and there was no detectable contribution to the phospholipid tail layer at neutral pH (S4B Table). Despite this, the thickness of PVL was approximately 67.6 ± 4.0 Å (S5D Fig and S4B Table). The data suggest limited complex formation without membrane penetration, consistent with the notion that PVL remained inactive under these conditions in the liposome leakage assay (Fig 1D). Together, these data demonstrate that PVL interacts with model membranes containing cardiolipin or PA at acidic pH, resulting in conformational changes and loss of membrane integrity, in the absence of proteinaceous receptors.

**PVL depends on sphingomyelin synthesis to kill macrophages.** The interaction of PVL with PA and cardiolipin under acidic conditions indicated that the leukocidin targets intracellular membranes, rather than the plasma membrane. We thus reasoned that PVL depends on additional factors besides cell surface receptors to kill host cells such as macrophages. To identify host genes critical for PVL-mediated cell death, we performed a genome-wide CRISPR-Cas9 based survival screen of PVL treated THP-1 differentiated macrophages [36]. The enrichment score of multiple gRNAs per gene after two independent PVL treatments compared to the untreated CRISPR library-transduced cells identified at least 25 gene candidates that confer susceptibility to the leukocidin (Fig 2A and S5 Table). As expected, the gene encoding C5aR1, *C5AR1*, was identified as one of the top hits, validating our approach (Fig 2A). In contrast to a recent screen in U937 monocytes engineered to express human C5aR1, we did not isolate the LukF-PV receptor, CD45, likely because CD45-deficient macrophages retain some sensitivity to PVL [12]. Genes encoding sphingomyelin synthase 1 (SGMS1) and solute carrier family 17 member 5 (SLC17A5) overlapped between the current and previous screen (Fig 2A) [12]. To

further validate the candidates, we used imaging-based analysis that enables quantification of macrophage cell death overtime, which demonstrated that targeting SGMS1 protected THP-1 differentiated macrophages from PVL cytotoxicity more so than Slc17A5 (S6A–S6J Fig and S6 Table). Targeting other top candidate genes reduced PVL cytotoxicity at varying levels overtime (S6A–S6J Fig). We therefore focused on SGMS1, particularly as it regulates lipid metabolism and may thus affect the interaction of PVL with host membranes.

To confirm a role of SGMS1 in PVL susceptibility, we generated two independent clones in THP-1 cells, that contained deletions or a single nucleotide insertion in the gRNA region (S7A Fig). SGMS1 transfers phosphocholine from PC to ceramide to form sphingomyelin (SM) and diacylgcerol (DAG) in a reversible manner, thus affecting SM but potentially other lipid levels. As expected, total SM and individual SM lipid species were significantly and markedly lower in both SGMS1-KO macrophage clones compared to Cas9-expressing THP-1 macrophages (Fig 2B and S7B Fig). Other major lipids, such as PC, ceramide, cholesterol, PA, phosphatidylethanolamine (PE), PS, phosphatidylinositol (PI), and CL remained relatively unchanged in SGMS1-KO macrophages, although lipid levels differed compared to WT cells to some degree (Fig 2B, S7B and S7C Fig). SGMS-1-deficient THP-1 monocytes were viable and differentiated to macrophages, which remained healthy (Fig 2C). The exposure of WT and SMGS1 KO THP-1 macrophages to the PVL subunits, LukS-PV and LukF-PV, did not affect cell viability (Fig 2C). In contrast to Cas9-expressing wild-type THP-1 macrophages, both SGMS1 KO clones remained viable after low and high concentrations of PVL (up to 2 µg/ml) for up to 5 h, to a similar degree as C5aR1-deficient macrophages (Fig 2C and S8A–S8D Fig). The SGMS1 KO clones remained susceptible to the leukocidin LukAB and nigericin, suggesting a toxin-specific effect (Fig 2C, S7D and S8E–S8H Fig). As sphingomyelin is associated with sterols in the plasma membrane, we next treated macrophages with the cholesterol-depleting agent, methyl-beta-cyclodextrin (MβCD). Importantly, MβCD treatment did not affect the C5aR1 surface expression and LukS-PV binding (Fig 2D and 2E). Despite this, MβCD protected THP-1 wild-type macrophages from PVL cytotoxicity, but not from LukAB (Fig 2F). We next determined PVL toxin activity using liposomes containing sphingomyelin and cholesterol. As expected, PC/sphingomyelin/cholesterol-containing liposomes were susceptible to control saponin treatment but remained refractory to PVL (Fig 2G), consistent with our previous evidence showing that PVL does not directly interact with sphingomyelin and cholesterol (Fig 1A). These results suggest that host-derived sphingomyelin and cholesterol play important roles in mediating PVL cytotoxicity, independent of directly interacting with PVL.

**Sphingomyelin and cholesterol promote retrograde trafficking C5aR1 and internalization of PVL.** Similar to cholesterol depletion, SGMS1 KO clones maintained C5aR1 and CD45 expression (Fig 3A and S9A Fig), as well as the LukAB receptor CD11b (S9A Fig). SGMS1 loss did result in lower levels of C5aR1 compared to WT cells (Fig 3A and 3B and S9B Fig). We did not detect upregulation of cell surface C5L2, a potential alternative receptor of LukS (S9A Fig). Consistent with these results, PVL or LukS-PV treatment of SGMS1 KO macrophages resulted in binding of LukS-PV to the cells, which was reduced compared to WT control cells but was significantly higher compared to C5aR1-deficient cells (Fig 3B and S9B Fig). LukS-PV binding to the cell surface of WT and SGMS1 KO macrophages was confirmed by indirect immunofluorescence staining and plasma membrane-specific probes (Fig 3C). The resistance of SGMS1 KO macrophages to even high concentrations of PVL indicated additional mechanisms besides lower levels of binding to the cell surface contribute to the phenotype. Surprisingly, when macrophages were treated with PVL, LukS-PV was also detected in intracellular puncta in WT cells within 5 min (Fig 3C). In contrast, LukS-PV remained primarily associated with the cell surface in SGMS1 KO macrophages up to 15 min (Fig 3C). PVL treatment of human induced pluripotent stem cell-derived macrophages (hiPSC-Mac) similarly resulted in LukS-PV staining of intracellular puncta within 5 and 15 min, which overlapped with C5aR1 signals (Fig 3D). LukS-PV and C5aR1 were not detected in PVL-treated C5aR1 KO hiPSC-Mac (Fig 3D). The C5aR1 receptor is internalized after C5a binding [37], but whether this depends on sphingolipid biosynthesis has not been addressed. C5a treatment caused the redistribution of C5aR1 to punctate structures in THP-1 macrophages, which was further increased in the presence of ammonium chloride by preventing lysosome acidification and thus C5aR1 degradation (S10 Fig) as reported previously [37]. In contrast, C5aR1 failed to localize to intracellular punctate structures

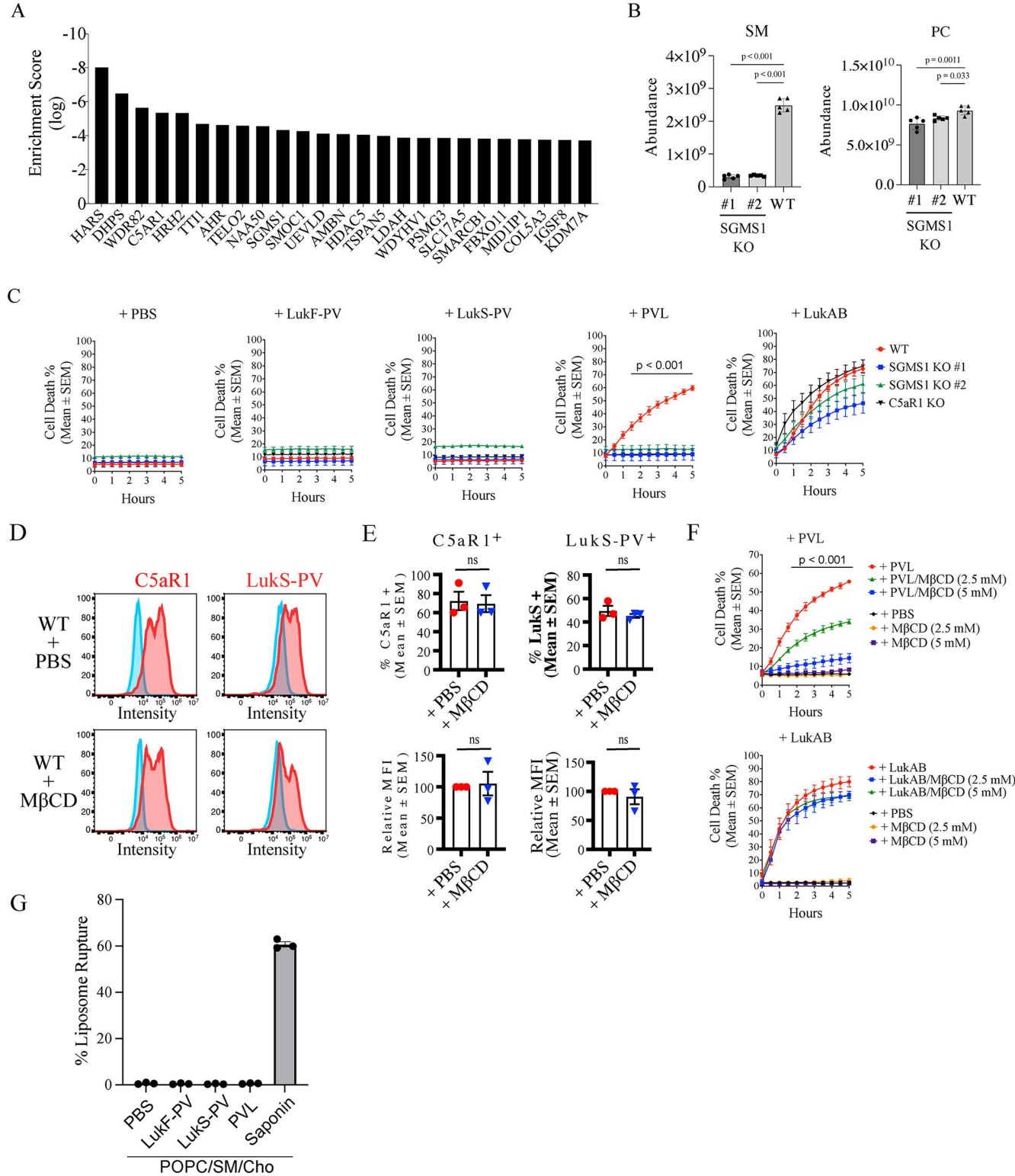

**Fig 2. PVL toxicity depends on host sphingomyelin and cholesterol. (A)** 25 most enriched genes identified in a CRISPR screen in human macrophages treated with PVL. **(B)** Relative abundance levels of total sphingomyelin (SM) and phosphatidylcholine (PC) in wild type and SGMS1 KO #1 and #2 THP-1 macrophages. Mean ± SEM from five independent experiments, *P* values (one-way ANOVA followed by Dunnett's multiple Comparison test)

are shown. **(C)** Cell death (Draq7-positive) of wild type and two independent SGMS1 KO THP-1 macrophages treated with PVL (62.5 ng/ml) or LukAB (15.6 ng/ml) overtime. Mean ± SEM from three independent experiments, P values (one-way ANOVA) are shown. **(D)** Flow cytometry of cell-surface levels of C5aR1 and LukS-PV in WT THP-1 macrophages treated with methyl-beta-cyclodextrin (MβCD) or control PBS. Blue line indicates isotype control, red line antigen-specific antibodies. Data representative of three independent experiments. **(E)** Mean fluorescence intensity (MFI) of C5aR1 and LukS-PV in PBS- or MβCD-treated macrophages. Mean ± SEM from three independent experiments. ns = not significant; unpaired t test. **(F)** Cell death (Draq7-positive) of human THP-1 macrophages treated with PVL (62.5 ng/ml) or LukAB (15.6 ng/ml) with PBS or MβCD (2.5 or 5 mM) overtime. Mean ± SEM from three independent experiments, P values (one-way ANOVA) are shown. **(G)** Sulforhodamine B encapsulated liposome of POPC/sphingomyelin/cholesterol (w/w 4:1:1) were treated with purified PVL (1 μg/ml), LukF-PV, LukS-PV, saponin, or control PBS and fluorescence determined relative to Triton X-100 treatment. Mean ± SEM from three independent experiments. The data underlying this Figure can be found in S1 Data.

in SGMS1 KO macrophages after C5a treatment (S10 Fig). Similar to sphingomyelin loss, depletion of cholesterol abrogated punctate staining of LukS-PV in PVL-treated macrophages despite normal levels of C5aR1 (Fig 3E). We next determined whether SGMS1 is also required during *S. aureus* infection, as previous studies have shown that *S. aureus* is internalized by macrophages and can survive intracellularly [38]. The absence of sphingomyelin synthesis, however, did not affect the phagocytosis of *S. aureus* and intracellular bacterial replication in THP-1 macrophages (Fig 3F). This is consistent with the finding that C5aR1 did not affect macrophage infection and intracellular replication of *S. aureus* (Fig 3F). However, both SGMS1 KO clones showed reduced cell death rates after 5 h post infections (Fig 3G). By that time, *S. aureus* had escaped from macrophages at considerable levels (S11 Fig). This suggests that *S. aureus* depends on host sphingomyelin synthesis to sufficiently kill macrophages after escape and that additional leukocidins are involved as macrophages deficient in C5aR1 were killed at similar rates as WT cells (Fig 3G). Taken together, sphingomyelin and cholesterol are required for PVL to redistribute from the plasma membrane to intracellular structures, similar to C5aR1.

## PVL affects lysosomal and mitochondrial health

Given the redistribution of PVL from the plasma membrane to intracellular puncta, we next determined whether the toxin affects the integrity of intracellular organelles, in addition to the plasma membrane. C5aR1 traffics to lysosomes after activation by C5a [37] and intracellular staining of C5aR1 after C5a treatment was dependent on SGMS1 (S10 Fig). Similarly, PVL treatment of hiPSC-Macs resulted in the redistribution of LukS-PV from the cell surface after 5 min to intracellular puncta that partially co-stained for the lysosomal protein LAMP-1 (Fig 4A). To address whether PVL affects the integrity of lysosomes, we performed live-cell imaging to assess lysosomal acidification using the fluorescent marker lysotracker red, together with the cell-impermeable nuclear stain, Draq7, to determine plasma membrane integrity. Untreated THP-1 macrophages remained viable (Draq7 negative) during the imaging, with little evidence of loss of lysosome integrity (lysotracker positive) (S12A Fig and S1 Video). After PVL treatment, wild-type THP-1 macrophages lost lysosomal staining which was followed by Draq7 staining, as quantitatively assessed at the single cell level over time (Fig 4B and S2 Video). In contrast, we did not detect any significant loss of lysosomal integrity in the SGMS1 KO clones treated with PVL (Fig 4B and S3 and S4 Videos). PVL treatment of WT THP-1 macrophages also led to the formation of galectin-3 puncta, which were largely absent in untreated control cells but readily induced after lysosome disruption with LLO-Me (S12B Fig). To assess whether lysosomal rupture by PVL contributes to macrophage killing, we used the lysosomal protease inhibitor, Ca-074 Me. Ca-074 Me reduced PVL-mediated cell death in human THP-1 macrophages, but did not prevent loss of lysosomal integrity (Fig 4C). As shown previously, PVL caused the activation of NLRP3 which contributed to macrophage death, but not loss of lysosomal integrity (Fig 4D). Taken together, the data show that PVL traffics to lysosomes of macrophages and causes loss of lysosomal integrity prior to plasma membrane rupture.

Given that PVL also interacted with cardiolipin and can traffic to mitochondria [39], we next used the fluorescent stain Tetramethylrhodamine (TMRM) to visualize mitochondrial membrane potential ($\Delta\Psi_m$) overtime using live-cell imaging. THP-1 macrophages rapidly lost its $\Delta\Psi_m$ within 2 h of PVL but not after control treatment, which preceded the loss of plasma membrane integrity (Fig 4E and S13A Fig). The $\Delta\Psi_m$ was unaffected in SGMS-1 KO macrophages treated with

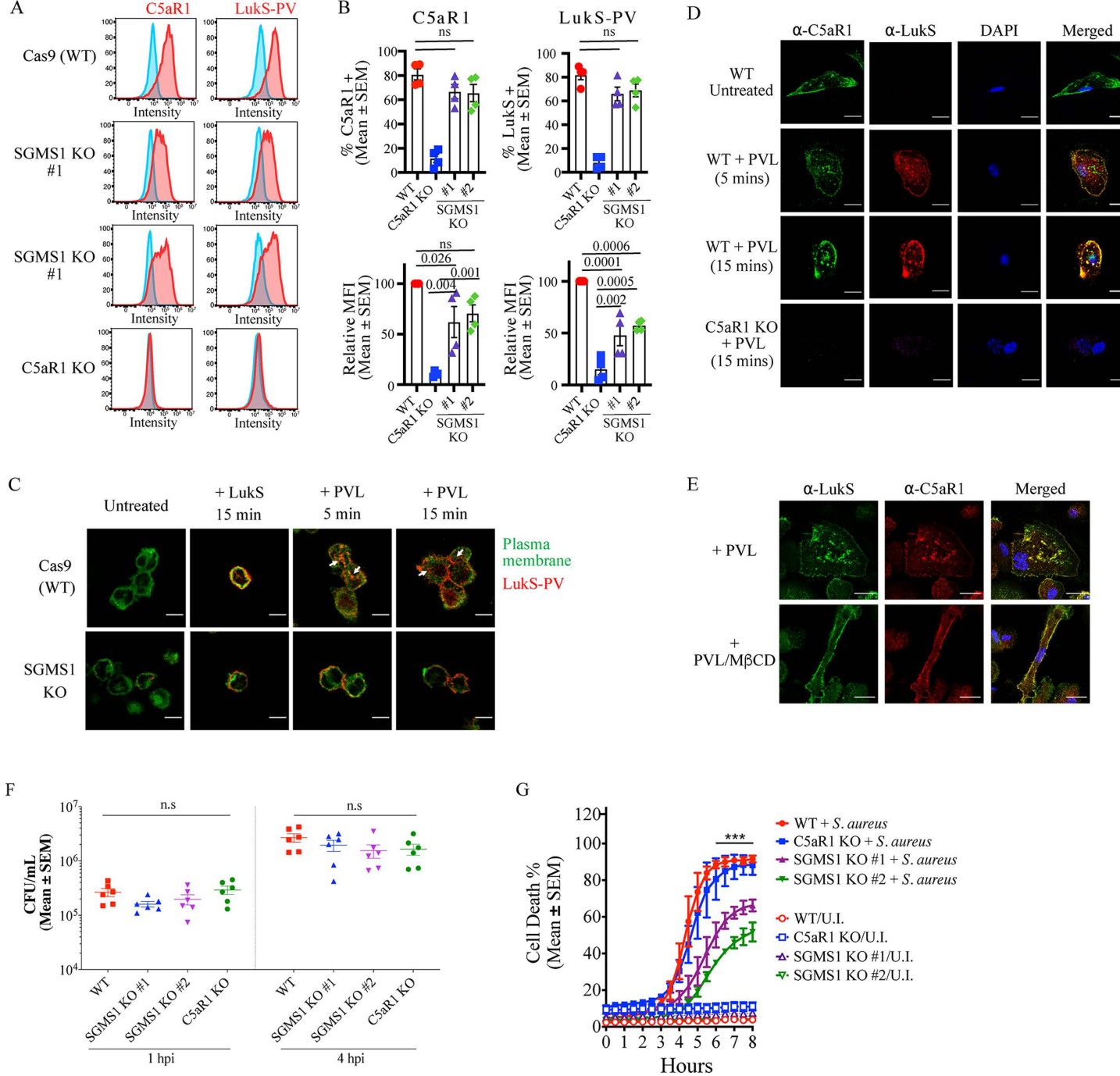

**Fig 3. PVL internalization in human macrophages based on sphingolipid and cholesterol. (A)** Cell-surface levels of C5aR1 and LukS-PV in WT Cas9 THP-1 macrophages, SGMS1 knockout, and C5aR1 knockout clones using flow cytometry. Blue line represents isotype control treatment, red line antigen-specific staining. Data representative of four independent experiments. **(B)** Flow cytometric analysis and mean fluorescence intensity (MFI) of C5aR1 and LukS-PV from panel A. Mean ± SEM from four independent experiments. ns = not significant; P values are shown by one-way ANOVA followed by Dunnett's multiple comparison test. **(C)** Localization of LukS-PV (red) in WT and SGMS1 KO macrophages treated with PVL (62.5 ng/ml) or LukS-PV (62.5 ng/ml) for 5 or 15 min. Plasma membrane was stained with DeepRed Cytopainter (green). White arrows indicate internalized LukS-PV signal. Scale bar is 20 μm. **(D)** WT and C5aR1-deficient human induced pluripotent stem cell (iPSC)-derived macrophages were treated with PVL (62.5 ng/ml; 5 or 15 min) and probed with anit-LukS-PV (red) and C5aR1 (green) antibodies, and nuclei stained with DAPI (blue). Scale bar is 20 μm. **(E)** Human iPSC-derived macrophages were treated with MβCD for 30 min prior to PVL exposure for 15 min. Cells were probed with anti-LukS-PV (green)

and C5aR1 (red) antibodies. Scale bar is 20 µm. **(F)** Colony forming unit (CFU) of *S. aureus* in WT, SGMS1 KO, and C5aR1 KO human THP-1 macrophages, infected at an MOI of 10, at 1 and 4 h post infection (hpi). Mean ± SEM from six independent experiments. One-way ANOVA showed no statistical significance (n.s.). **(G)** Cell death (Draq7-positive) of WT, SGMS1 KO, and C5aR1 KO THP-1 macrophages after infection with *S. aureus*. Mean and ± SEM from three independent experiments. *** indicate *P*-value <0.01 (one-way ANOVA between WT and SGMS1 KO). The data underlying this Figure can be found in S1 Data.

PVL (Fig 4E). Consistent with the notion that PVL compromised mitochondrial membranes, PVL-treated THP-1 macrophages released the soluble mitochondrial protein Smac/DIABLO, but not membrane bound VDAC, to the cytosol in a time-dependent manner, and as early as 60 min post-treatment (S13B Fig). These results suggest that PVL also causes loss of mitochondrial integrity preceding plasma membrane rupture in THP-1 macrophages.

### *S. aureus* infection disrupt lysosomal health in macrophages

*S. aureus* is able to survive phagocytosis by macrophages by escaping and killing host cells. The underlying mechanisms remain controversial, as intracellular *S. aureus* replication has been noted to depend on acidic organelles, such as lysosomes [40–42]. On the other hand, intracellular *S. aureus* replication has also been shown to depend on the escape to the cytosol of macrophages [40,43]. Rather than using predefined time points, we next followed intracellular GFP-expressing *S. aureus* by applying live-cell imaging. We also monitored organellar acidification using lysotracker red staining and loss of plasma membrane integrity (i.e., cell death) with a DNA stain (S5 Video). Uninfected THP-1 macrophages remained viable over extended periods of time with minimal loss of acidification (Fig 5). In contrast, intracellular GFP-expressing *S. aureus* caused macrophage death within 5 h (Fig 5). Initially, GFP-expressing *S. aureus* resided within acidic organelles, which subsequently lost lysotracker staining within 50 min (Fig 5). *S. aureus* continued to replicate within defined regions of macrophages, which lost all lysotracker staining by 80 min post infections (Fig 5). Only then, macrophages died as indicated by DNA staining, harboring increased numbers of GFP-expressing *S. aureus* within defined subcompartments (Fig 5). Over time, GFP-expressing *S. aureus* escaped from the subcompartments and entered a rapid growth phase until macrophages were completely filled with bacteria (Fig 5). By 220 min, bacteria escaped macrophages and continued to expand extracellularly (Fig 5). Taken together, this demonstrates that *S. aureus* replicates within acidic and not-acidic organelles, the cytosol, and extracellularly during macrophage infections. This raises the prospect that host and bacterial factors affect the different intracellular *S. aureus* stages.

### Targeting lysosomal acidification affect PVL toxicity and *S. aureus* macrophage infections

Given that the above data showed that PVL is active within acidic organelles and *S. aureus* escapes from macrophages, we next wanted to determine whether PVL contributes to *S. aureus* infections in macrophages. To functionally test whether acidic environments are important for PVL activity, we treated macrophages with the membrane-associated vacuolar ATPase inhibitor Bafilomycin A1, which prevents acidification of lysosomes. Bafilomycin A1, but not vehicle control treatment, impaired the ability of recombinant PVL, but not nigericin, to kill THP-1 macrophages (Fig 6A and S14A Fig) and reduced galectin-3 puncta formation in PVL-treated macrophages (S12B Fig). Next, we pretreated THP-1 macrophages with Bafilomycin A1, then infected with WT and Δ*pvl*, and assessed bacterial loads in macrophages using antibiotic protection assays at 1 and 4 h post infections. The time points correspond to early and late-stage intracellular replicative phases of *S. aureus*, respectively (Fig 5). Under the condition used, both WT and Δ*pvl* infected THP-1 macrophages and established intracellular infections at similar rates as indicated by similar colony-forming units (CFUs) at 1 h post infection (Fig 6B). By 4 h post infection, intracellular *S. aureus* numbers were increased in both WT and Δ*pvl* infected macrophages, which were significantly reduced in the presence of Bafilomycin A1 (Fig 6B). We also assessed extracellular bacterial burdens in macrophage-free supernatants, indicative of *S. aureus* escape from macrophages. *S. aureus*

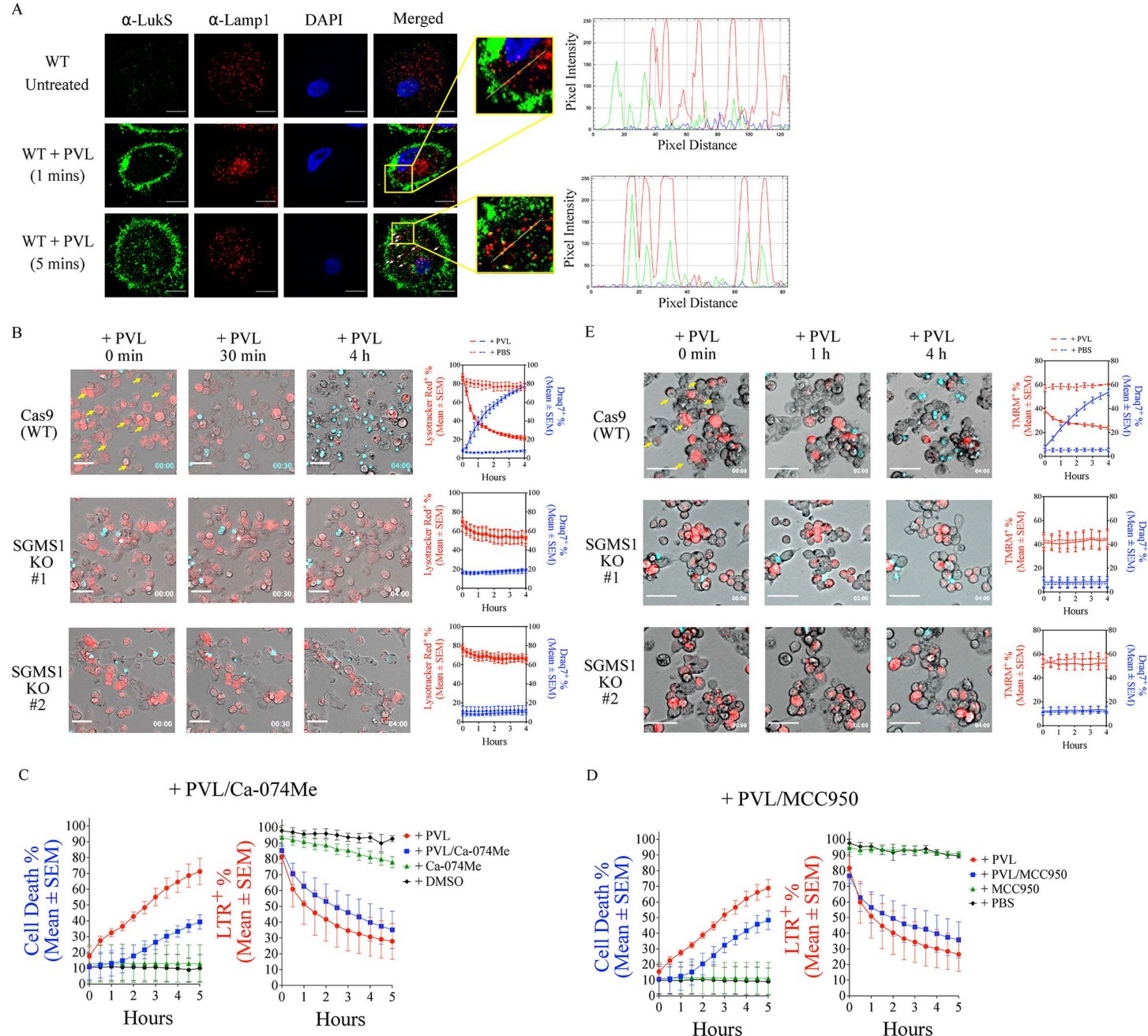

**Fig 4. PVL triggers lysosome and mitochondria permeabilization in human macrophages.** **(A)** PVL (62.5 ng/ml)-treated wild type (WT) human iPSC-derived macrophages were probed with anti-LukS-PV (green) and LAMP1 (red) antibodies, nuclei stained with DAPI (blue) at 1 and 5 min post-treatment. White arrows show co-localization of LukS-PV and LAMP1. Scale bar is 20 μm. The yellow line was used to generate pixel intensity and distance of LukS-PV (green) and LAMP1 (red). **(B)** WT and SGMS1 KO THP-1 macrophages were treated with PVL (62.5 ng/ml) and LysoTracker Red (red, acidic organelles) and Draq7 (blue, dead cells) fluorescence imaged over time. Yellow arrows indicate cells that loose LysoTracker Red signals prior to Draq7. Scale bar is 50 μm. Quantification of fluorescence intensity of LysoTracker Red and Draq7 in PVL and PBS control cells. Mean ± SEM from three independent experiments shown. **(C)** Quantification of Draq7 (dead cells, left panel) and LysoTracker Red staining (LTR, acidic organelles, right panel) of THP-1 macrophages exposed to PVL (62.5 ng/ml) and Ca-074Me or DMSO control overtime. Mean ± SEM from three independent experiments. ***$P < 0.001$, ns = not significant (one-way ANOVA). **(D)** Cell death (Draq7-positive, left panel) and LysoTracker Red (LTR, right panel) staining of THP-1 macrophages exposed to PVL and MCC950 or DMSO control overtime. Mean ± SEM from three independent experiments. **$P < 0.01$, ns = not significant (one-way ANOVA). **(E)** WT and SGMS1 KO THP-1 macrophages were treated with PVL (62.5 ng/ml) for the indicated time and MitoTracker

TMRM (red) and Draq7 (blue) fluorescence imaged. Yellow arrows indicate the loss of TMRM signals prior to Draq7 uptake. Scale bar is 50 μm. Quantification of fluorescence intensity of TMRM (red) and Draq7 (blue) in PVL and PBS control cells. Mean ± SEM from three independent experiments shown. The data underlying this Figure can be found in S1 Data.

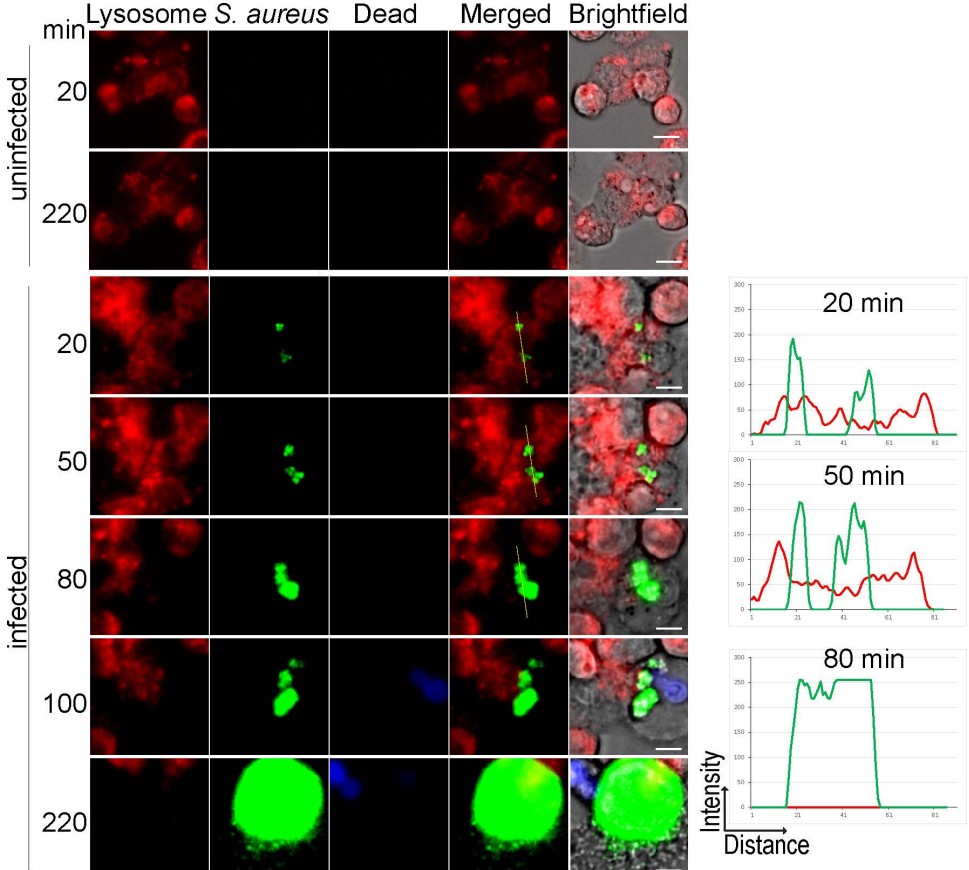

**Fig 5. Intracellular replication of *S. aureus* in human macrophages.** THP-1 macrophages were stained with LysoTracker red (Lysosome) and infected with or without GFP-expressing *S. aureus* (MOI = 10) in the presence of Draq7 (dead macrophages). Data represents images from live-cell microscopy at indicated time points from three independent experiments. Pixel intensity and distance of LysoTracker (red) and GFP (green) along the yellow line are shown on the right. Scale bare = 10 μm. The data underlying this Figure can be found in S1 Data.

escaped from macrophages by 4 h post infections, whereby the extracellular burden was significantly lower after Δ*pvl* infections compared to WT and complemented strains (Fig 6C). Bafilomycin A1 treatment significantly reduced extracellular burdens in WT and complemented strain-infected macrophages, and also to some degree in Δ*pvl* infections although this was not statistically significant (Fig 6C). Reduced extracellular Δ*pvl* burdens were associated with lower rates of bacterial escape from macrophages as determined by live-cell imaging (Fig 6D and S6, S7 and S8 Videos) and macrophage cell death rates (Fig 6E) compared to WT and complemented strains. Bafilomycin A1 treatment reduced escape of WT and complemented strains, but also Δ*pvl* from infected macrophages (Fig 6D). Bafilomycin A1 also impaired escape of RFP-expressing WT *S. aureus* from macrophages (S14B Fig). Bafilomycin A1 treatment reduced macrophage cell death rates compared to the control group in WT and Δ*pvl* infections (Fig 6F and 6G and S9 Fig and S10 Videos). Bafilomycin A1 treatment did not affect the level of PVL (LukS-PV) in macrophages infected with WT and the complemented

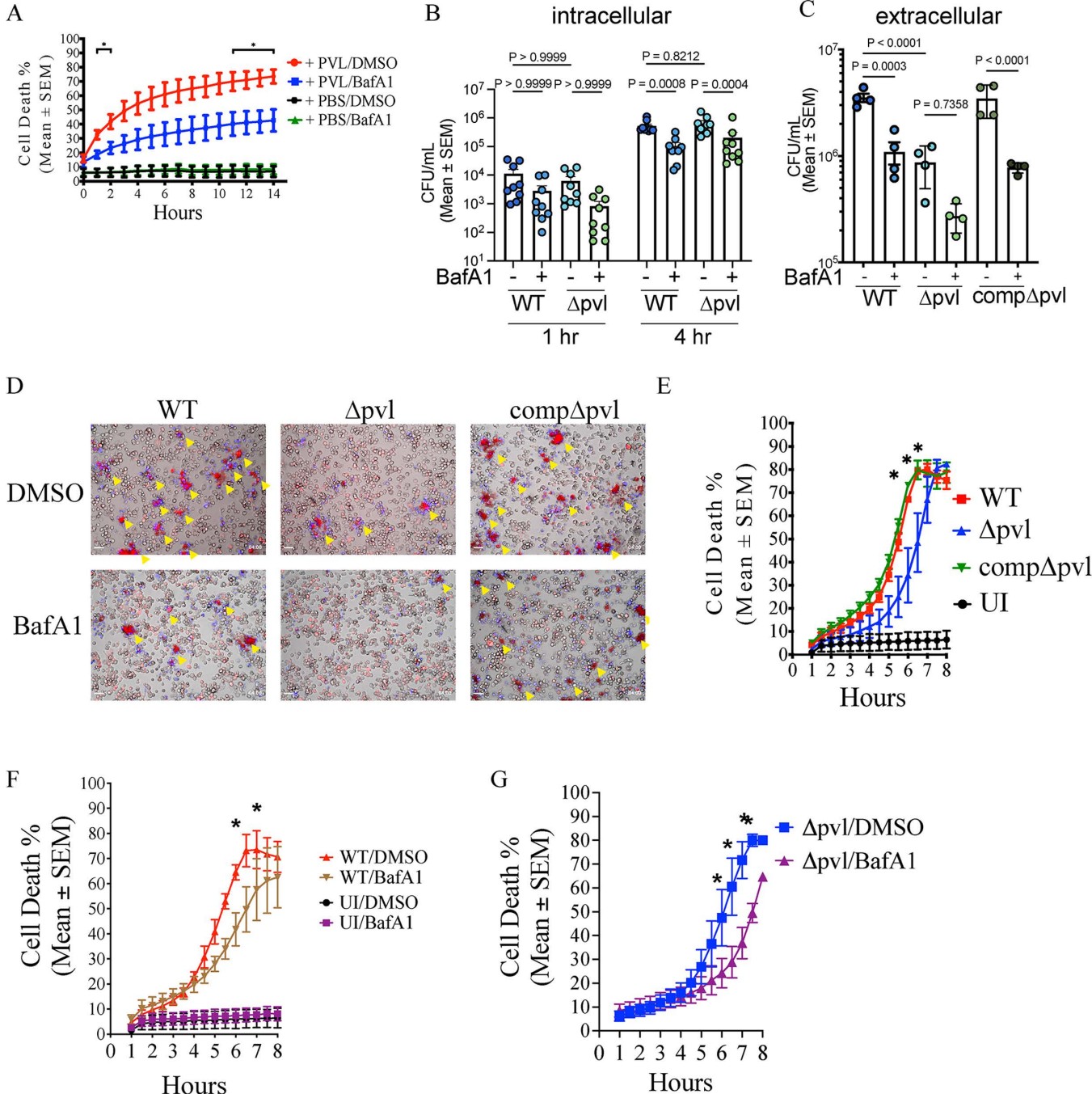

**Fig 6. PVL and lysosomal acidification promotes *S. aureus* escape from human macrophages. (A)** Cell death (Draq7-positive) of THP-1 macrophages treated with PVL (62.5 ng/ml) and Bafilomycin A1 (BafA1, 100 nM) or DMSO control overtime. Mean ± SEM from three independent experiments. *P < 0.05 for PVL/DMSO vs. PVL/BafA1 at 1, 2, 11−14 h post toxin treatment; by unpaired *t* test. **(B)** Lysostaphin protection assay whereby THP-1 macrophages were infected with *S. aureus* WT and Δ*pvl* (MOI = 10) and colony forming unit (CFU) determined at 1 h and 4 h post infection. Intracellular CFUs were derived from macrophages. Mean ± SEM from nine independent experiments shown. *P*-values from one-way ANOVA. **(C)** THP-1 macrophages were infected with WT, Δ*pvl*, and complemented strain (compΔ*pvl*) (MOI = 10) and extracellular bacterial numbers (CFU) determined in the supernatant after 4 h. Mean ± SEM from four independent experiments shown. *P*-values from one-way ANOVA. **(D)** THP-1 macrophages infected with WT, Δ*pvl*, and compΔ*pvl* *S. aureus* in the presence of MitoTracker TMRM (healthy mitochondria, bacteria, red) and Draq7 (dead macrophages, blue) were imaged with live-cell microscopy. Images taken at 4 h post infection. Yellow arrows indicate escaped intracellular bacteria stained with TMRM.

Scale bar is 50 µm; Images representative of three independent experiments. **(E)** Cell death (Draq7 positive) of THP-1 macrophages infected with WT, Δ*pvl* and compΔ*pvl S. aureus.* Mean ± SEM from four independent experiments. * indicates *P*-value <0.01 (two-way ANOVA). **(F, G)** Cell death (Draq7 positive) of THP-1 macrophages infected with **(F)** WT or **(G)** Δ*pvl S. aureus* and treated with BafA1 or DMSO. Mean ± SEM from four independent experiments. * indicates *P*-value <0.01 (two-way ANOVA). The data underlying this Figure can be found in S1 Data.

strain, nor *in vitro* growth rates of *S. aureus* strains (S14C–S14F Fig). Interestingly, α-hemolysin levels were increased in Bafilomycin A1 treated macrophages infected with WT, Δ*pvl* and complemented strains (S14C Fig). Together, these data indicate that endo/lysosomal acidification promotes host cell death and *S. aureus* escape from macrophages, which are accelerated in the presence of PVL.

## Discussion

The *S. aureus* leucocidin PVL is thought to kill macrophages and other phagocytes by forming pores within the plasma membrane depending on cell surface receptors such as C5aR1. The initial binding of the subunit LukS-PV to C5aR1, however, does not necessarily lead to pore formation [44], and imaging PVL complexes and detecting toxin activity have been challenging [45]. We now identify here that leucocidin activity depends on host sphingomyelin biosynthesis. Rather than enabling the interaction with cell surface receptors or the plasma membrane, we show that sphingomyelin synthesis promotes internalization of C5aR1 and PVL to acidic organelles, such as lysosomes. Using model membranes, we further demonstrate that recombinant PVL directly interacts with the phospholipids PA and cardiolipin but only at acidic pH, which are enriched in lysosomal and mitochondrial compartments, respectively. PVL is able to directly interact and compromise the integrity of model lipid bilayers that mimic lysosomal and mitochondrial environments, as determined by QCM-D, neutron reflectometry, and functional assays. Thus, we propose an alternative model whereby C5aR1 targets PVL to intracellular compartments where under acidic conditions the leucocidin interacts with anionic phospholipids, which is sufficient for pore formation. Consequently, PVL primarily disrupts lysosomes and mitochondria prior to plasma membrane rupture.

Bicomponent leukocidins such as PVL are proposed to assemble into a membrane pore-forming complex after the initial interaction of the soluble subunits with their respective cell surface receptors [3]. While C5aR1 promotes binding of PVL to host cells, we now demonstrate that the host cell receptor also enables intracellular trafficking of the leucocidin, which is essential for the toxin to kill macrophages. The internalization of C5aR1 and PVL is dependent on sphingomyelin and cholesterol, which has not been appreciated previously. PVL may not directly bind these particular lipids, as is known with other bacterial toxins [8]. Rather, it is well established that cell surface receptors, such as C5aR1, are rapidly internalized after binding their cognate ligands [46–50]. Internalization of cell surface receptors to acidic organelles is thought to control their activity by promoting dissociation of the ligand and recycling of the receptor to the cell surface or degradation via lysosomes [37]. While the rapid internalization and degradation may potentially protect host cells from toxins that bind C5aR1, our data suggest that trafficking to endo/lysosomal compartments promotes PVL activity. This is supported by the fact that PVL as detected by the LukS-PV subunit still binds to SGMS1-deficient macrophages, albeit at lower levels, but fails to kill. Alternatively, SGMS1-deficient macrophages may be inherently resistant to PVL killing as sphingomyelin and downstream metabolites such a ceramide promote cell death signaling [51,52]. This is because PVL exposure leads to the activation of the NLRP3 inflammasome, which triggers pyroptotic cell death in macrophage [14,17]. However, there is no marked difference in macrophage cell death rates between WT and SGMS1-deficient macrophages treated with the NLRP3 activator nigericin. We also note ~50% reduction of LukS-PV binding capacity to SGMS1-deficient macrophages treated with PVL. The underlying mechanism, which remains unknown, for the reduced PVL interaction with SGMS1-deficient macrophages may also contribute to the resistance of these cells to the toxin. It should be noted, however, that SGMS1-deficient macrophages remain resistant to high concentrations of PVL. Compromising the acidification of lysosomes with Bafilomycin A1 reduces the ability of recombinant PVL to kill macrophages. This likely also applies to human neutrophils [45] and in *S. aureus* infected host cells [53]. Thus, preventing PVL internalization and activation may lead to

alternative strategies aimed at neutralizing toxins to improve outcomes of *S. aureus* infections [54–56]. Intriguingly, statin treatments to lower cholesterol levels are associated with reduced *S. aureus* infections [57]. Whether this is linked directly to reduced PVL internalization as shown here awaits to be seen.

Structural studies on γ-hemolysin suggest that oligomerization of the leucocidin subunits after the interaction with the host receptors triggers conformational changes and the insertion of a hydrophobic region into host membranes [10,11]. Based on this, the current model proposes that the host receptors not only concentrate PVL at the plasma membrane but also enable membrane interaction and pore formation. Our data now suggest that additional mechanisms are at play that unleash PVL activity. In particular, PVL is able to directly bind and damage model membranes independent of C5aR1 or CD45, as long as the lipid bilayers contained PA and CL under acidic rather than neutral pH conditions. PA and CL are considered anionic lipids and may thus enable electrostatic interactions with amino acids of either LukF or LukS-PV, although at low pH these interactions are likely compromised [58]. Both PA and CL contain a small polar head group relative to the size of the tails that is likely to form cone-shaped lipids [59], whereas the anionic PS is a cylindrical shape lipid [60]. This would indicate that the anionic, cone-shaped lipids may promote the interaction of subunits, and promote the oligomerization process of PVL complex. Before the discovery of protein receptors, biophysical studies indicated that γ-hemolysin and its LukF subunit can bind PC, the major lipid of mammalian membranes [61]. In contrast to other toxins, demonstrating a direct interaction of PVL with model membranes, however, has remained challenging [61]. Our discovery that specific lipids and pH affect PVL-membrane interactions will enable further investigations into how the toxin penetrates membranes.

The *S. aureus* strain USA300 has gained the ability to survive and replicate within macrophage phagolysosomes [41,62]. Except for *Coxiella* and *Leishmania* parasites, most other pathogens evade lysosomes in macrophages by preventing the fusion of phagosomes with lysosomes or escaping to the cytosol [63]. Paradoxically, *S. aureus* USA300 appears to depend on an acidic environment for intracellular survival as Bafilomycin A1 treatment reduced intracellular *S. aureus* numbers as shown here and previously [40]. It is tempting to speculate that the dependency on acidic environments within macrophages may relate to the observed activity of recombinant PVL under acidic conditions. Based on the data presented here, we propose that *S. aureus* replicates within acidic organelles, but only transiently. There is detectable bacterial replication within non-acidified organelles, potentially within neutralized lysosomes. Notwithstanding other mechanisms, PVL may rapidly disrupt lysosomal/endosomal membranes by directly engaging with lipids such as PA to form pores under acidic conditions, which neutralizes the environment. Eventually, lysosome rupture promotes escape of *S. aureus* to the cytosol and rapid bacterial growth. It is evident from our data, however, that *S. aureus* is still able to replicate in macrophages treated with Bafilomycin A1 and/or in the absence of PVL, suggesting that *S. aureus* is able to escape from lysosomes via additional mechanisms. While PVL is thought to promote intracellular survival of *S. aureus,* additional toxins contribute during these interactions [53,64]. We also note that Bafilomycin A1 treatment triggers the upregulation of α-hemolysin in infected macrophages, suggesting that *S. aureus* is able to respond to changing conditions accordingly. Whether the activity of other toxins, including leukocidins, is pH sensitive awaits to be seen.

After successful intracellular replication, *S. aureus* escapes from macrophages. PVL expression increases extracellular bacterial numbers after macrophage infections. This correlates with increased macrophage cell death rates, rather than with intracellular burdens in Δ*pvl* infections. Given that *S. aureus* initially escapes into the cytosol of macrophages, PVL is likely able to target other membranes and organelles, such as mitochondria. These additional interactions would depend on host lipids rather than cell surface receptors. PVL was able to target and damage model membranes containing the mitochondrial lipid, cardiolipin. Intriguingly, PVL targets mitochondria in living cells and translocates into isolated mitochondria, likely enabling binding to cardiolipin within the inner mitochondrial membrane which is thought to be exposed to slightly more acidic conditions compared to the cytosol [65,66]. As observed with the loss of lysosome integrity, recombinant PVL also causes the rapid decline of mitochondrial health as determined by its membrane potential. Whether the loss of mitochondrial health is mediated directly by PVL or by proteases released from damaged lysosomes, as was described

recently, awaits to be seen [67]. In contrast to mutants deficient in global regulation of virulence factors, PVL only contributes to the escape from macrophages to some degree (2–3 fold difference in CFU during the early phase of escape) [53,68]. Thus, capturing the contribution of individual bacterial factors during these host-pathogen interactions depends on sensitive assays. Determining bacterial numbers (i.e. CFUs) in macrophages and their supernatants (intracellular versus extracellular) is commonly used to assess host cell escape, although dead host cells detach and may rupture during handling, and bacteria adhere to culture vessels. Live-cell imaging further suggests that the escape from macrophages is promoted by PVL and acidic organelles. It is also clear that *S. aureus* readily overwhelms macrophage cultures, regardless of PVL and/or Bafilomycin A1 treatment under these conditions. Surprisingly, bacterial escape and macrophage cell death rates are also affected by Bafilomycin A1 treated in Δ*pvl* infected macrophages compared to control groups. Previous studies have shown that acidic environments within macrophages activate signaling pathways in *S. aureus* that promote intracellular replication, independent of toxin production [42]. This suggests that PVL and additional pathways enable *S. aureus* to escape from macrophages in a timely manner. It is formally possible that lysosomal independent effects of Bafilomycin A1 are at play. Alternatively, PVL contributes to *S. aureus* escape depending on neutral, rather than acidic, pH conditions. Finally, cathepsins released by damaged lysosomes may activate host cell death factors, such as NLRP3, which contribute to *S. aureus* escape from macrophages [53]. Thus, further studies are required to fully assess how Bafilomycin A1 affects *S. aureus* infection in macrophages and the role of PVL and other toxins during escape.

Limitations of this study. We have discovered that PVL can directly interact and damage model membranes depending on the binding to anionic lipids that are enriched in intracellular membranes. The activity of PVL on model membranes, however, was only detectable using high concentrations (1–100 µg/ml). In clinical samples, the PVL concentration ranges from low ng/ml to 30 µg/ml, and can reach >100 µg/ml in rare cases depending on the infection site [69]. As shown here, cultured human macrophages are highly sensitive to PVL, whereby low ng/ml concentrations are sufficient for rapid killing. It thus remains unclear whether the PVL-lipid interactions occur in macrophages and in other susceptible innate immune cells *in vivo*, or whether they apply to non-phagocytic cells that are deficient in C5aR1 and CD45 and thus require high toxin concentrations. It awaits to been seen whether the identified interactions between PVL and host lipids occur under physiological conditions and whether they contribute to *S. aureus*-associated diseases. To visualize these interactions in living cells and animals, and whether they contribute to PVL activity in model infections depends on advanced tools. This is because depleting host lipids can be challenging due to their essential roles and redundant metabolic pathways. Similarly, the current mice infection models do not recapitulate all leukocidin–host cell interactions as PVL only interacts with human but not mouse receptors. Thus, future studies are aimed to address whether the interference with leukocidin activity in infections by repurposing drugs that alter PVL trafficking and/or organelle function prevent *S. aureus* infections and promote bacterial clearance.

## Supporting information

**S1 Fig. QCM-D profiles.** QCM-D profiles for purified PVL in different pH buffers on supported lipid bilayer made of **(A)** POPC or **(B)** POPC/POPS (w/w 2:1). Liposome addition indicated with 1↓, buffer exchange with 2↓ and PVL treatment (100 µg/ml) with 3↓. Blue: frequency; red: dissipation. Data representative of two independent experiments. **(C)** Mitochondria isolated from *S. cerevisiae* were treated with/without proteinase K after import of LukS-PV for indicated time. Mitochondrial extracts were probed with antibodies against LukS-PV, Tom70, and Tim23. The data underlying this Figure can be found in S1 Data.
(TIF)

**S2 Fig. PVL ruptures liposomes containing anionic phospholipids at acidic pH. (A–B)** Sulforhodamine B encapsulated liposomes of **(A)** POPC, **(B)** POPC/TOCL (w/w 2:1) or **(C)** POPC/POPA (w/w 3:1) were treated with purified PVL and its subunits LukS-PV and LukF-PV (1 µg/ml) in neutral or acidic buffers overtime. Increased sulforhodamine B

fluorescence due to its release was determined overtime, compared to Triton X-100 treatment. PVL and subunits were added at the indicated time point (1↓). Mean from three independent experiments are shown. The data underlying this Figure can be found in S1 Data.
(TIFF)

**S3 Fig. Neutron reflectometry of lipid bilayers. (A–D)** Neutron reflectometry (NR) analysis of distinct lipid bilayers in the absence of PVL. **(A–B)** NR analysis of POPC/TOCL (w/w 2:1) membrane in **(A)** pH 7.4 or **(B)** pH 5.0 buffer. **(C–D)** NR profiles for POPC/POPA (w/w 2:1) membrane in **(C)** pH 7.4 or **(D)** pH 5.0 buffer. Black curve: measurement in $D_2O$ buffer; Red curve: measurement in contrast matched Si buffer (CmSi); Blue curve: measurement in $H_2O$ buffer. **(E–H)** Scattering length density profiles with cartoons for **(E–F)** POPC/TOCL at **(E)** pH 7.4 or **(F)** pH 5.0 buffer, and **(G–H)** POPC/POPA membrane at **(G)** pH 7.4 or **(H)** pH 5.0 buffer. Black line: measurement in $D_2O$ buffer; Red line: measurement in contrast matched Si buffer (CmSi); Blue line: measurement in $H_2O$ buffer. The data underlying this Figure can be found in S1 Data.
(TIFF)

**S4 Fig. PVL and cardiolipin containing membranes. (A–D)** Neutron reflectometry (NR) analysis of lipid bilayers with or without purified PVL (100 μg/ml). **(A–B)** POPC/TOCL (w/w 2:1) membrane in $D_2O$ buffer at **(A)** pH 7.4 or **(B)** pH 5.0 with (red curve) or without PVL (blue curve). **(C–D)** POPC/POPA (w/w 2:1) membrane in $D_2O$ buffer at **(C)** pH 7.4 or **(D)** pH 5.0 with (red curve) or without PVL (blue curve). **(E–F)** NR profiles for purified PVL interacting with **(E)** POPC/TOCL (w/w 2:1) and **(F)** POPC/POPA (w/w 2:1) membrane in $D_2O$ buffer at pH 5.0 (red curve) or 7.4 (blue curve). The data underlying this Figure can be found in S1 Data.
(TIFF)

**S5 Fig. PVL does not form a protein complex on supported lipid bilayers containing anionic phospholipids at neutral pH. (A–B)** Neutron reflectometry (NR) analysis of purified PVL (100 μg/ml) interacting with **(A)** POPC/TOCL (w/w 2:1) and **(B)** POPC/POPA (w/w 2:1) membrane in pH 7.4 buffer. Black curve: measurement in $D_2O$ buffer; Red curve: measurement in contrast matched Si buffer (CmSi); Blue curve: measurement in $H_2O$ buffer. **(C–D)** Scattering length density profiles with cartoons for PVL with **(C)** POPC/TOCL and **(D)** POPC/POPA membrane. Black line: measurement in $D_2O$ buffer; Red line: measurement in contrast matched Si buffer (CmSi); Blue line: measurement in $H_2O$ buffer. The data underlying this Figure can be found in S1 Data.
(TIFF)

**S6 Fig. Verification of 10 gene candidates of a genome-wide CRISPR screen using live-cell imaging.** Cell death (Draq7-positive) of wild type, homogenous C5aR1 knockout and 4 independent heterogenous THP-1 macrophages targeting **(A)** AHR, **(B)** SGMS1, **(C)** HDAC5, **(D)** SLC17A5, **(E)** SMARCB1, **(F)** FBXO11, **(G)** MID1IP1, **(H)** IGSF8, **(I)** LAMTOR2, and **(J)** TTC14 treated with PVL overtime. The data underlying this Figure can be found in S1 Data.
(TIFF)

**S7 Fig. Characterization of SGMS1 KO THP-1 macrophages. (A)** Genomic sequence of sgRNA targeting site for two independent clones of SGMS1 knockout THP-1 cells. **(B–C)** Clustered heatmap **(B)** and relative abundance levels **(C)** of lipids in wild type and SGMS1 KO clones #1 and #2. Phosphatidic acid (PA), phosphatidylethanolamine (PE), phosphatidylserine (PS), phosphatidylinositol (PI) and cardiolipin (CL). **(D)** Cell death (Draq7-positive) of Cas9-expressing WT or SGMS1 knockout THP-1 macrophages treated with nigericin or DMSO overtime. Mean ± SEM from three independent experiments shown. The data underlying this Figure can be found in S1 Data.
(TIFF)

**S8 Fig. Dose-response of purified PVL and LukAB on human THP-1 macrophages. (A–H)** Cell death (Draq7-positive) of wild type, SGMS1 KO or C5aR1 KO THP-1 macrophages treated with different concentrations of **(A–D)** purified

recombinant PVL (32, 62.5, 500, 2,000 ng/ml) or **(E–H)** purified recombinant LukAB (3.9, 7.8, 15.6, or 31.3 ng/ml) overtime. Mean ± SEM from three independent experiments shown. The data underlying this Figure can be found in S1 Data. (TIFF)

**S9 Fig. Analysis of SGMS1-deficient macrophages. (A)** Cell-surface levels of CD45, CD11b and C5L2 (C5aR2) in Cas9-expressing THP-1 derived macrophages and two independent clones of SGMS1 knockout macrophages using flow cytometry. Data representative of three independent experiments. **(B)** Total cell lysates of WT, C5aR1 and SGMS1 KO THP-1 macrophages treated with PVL for 5, 15, and 30 min were probed with antibodies against LukS-PV and C5aR1. Untreated (UT) and recombinant LukF-PV and LukS-PV were included as controls. Ponceau staining was used as loading control. Data representative of three independent experiments. (TIFF)

**S10 Fig. C5aR1 internalization requires sphingomyelin in THP-1 macrophages.** THP-1 macrophages were pre-treated with cycloheximide with/without ammonium chloride for 1 h prior to C5a treatment for 3 h and then probed with antibody against C5aR1 (magenta). Nuclei are stained with DAPI (blue). Scale bar is 20 μm. (TIFF)

**S11 Fig. Role of sphingomyelin in *S. aureus* infection.** Images from live-cell video microscopy of WT, C5aR1 and SGMS KO THP-1 macrophages infected with DsRed-expressing *S. aureus* (USA300, SF8300 clone, red) in the presence of Draq7 (blue, dead cells) at 2 and 5 h post infection. Scale bar is 50 μm; Images representative of three independent experiments. (TIF)

**S12 Fig. PVL ruptures lysosomes. (A)** Lysotracker Red stained wild type or SGMS1 KO THP-1 macrophages were treated with PBS in the presence of Draq7 (blue, dead cells) for indicated time. Scale bar is 50 μm; Data representative of three independent experiments. **(B)** Galectin-3 staining (red) of THP-1 macrophages treated with LLO-Me (500 μM), PVL (62.5 ng/ml) or control vehicle with or without Bafilomycin A1 (BafA, 100 nM) for 1 h. Nuclei were stained with DAPI (blue). Scale bar is 20 μm. Cells containing Galectin-3 puncta (% Gal3 puncta) were quantified from >50 individual cells per experiments. Mean, SEM, and *P*-values from three independent experiments shown. The data underlying this Figure can be found in S1 Data. (TIF)

**S13 Fig. PVL targets and damages mitochondria. (A)** Single-cell analysis of MitoTracker TMRM stained (red) wild type (Cas9 WT) or SGMS1 KO THP-1 macrophages treated with PBS in the presence of Draq7 (blue, dead cells) for indicated time. Scale bar is 50 μm; Data representative of three independent experiments. **(B)** THP-1 macrophages were treated with PVL for the indicated times, subunit LukS-PV, LukF-PV, and the apoptosis inducing compounds staurosporine (STS) and cycloheximide (CHX) for 6 h or left untreated (UT). The cytosolic and mitochondrial fractions were probed with antibodies against Smac/DIABLO, VDAC (mitochondrial protein) and tubulin (loading control). (TIFF)

**S14 Fig. Targeting lysosomes in PVL-treated macrophages. (A)** Cell death (Draq7-positive) of THP-1 macrophages treated with nigericin (Nig), and Bafilomycin A1 (BafA1) or DMSO overtime. Mean +/- SEM from three independent experiments shown. **(B)** THP-1 macrophages infected with DsRed-expressing WT *S. aureus* (MOI = 10) in the presence of Bafilomycin A1 (BafA1) or vehicle control were analyzed by time-lapse imaging. Images are from 2 and 5 h post infection. Draq7 staining was included to identify dead cells. **(C)** THP-1 macrophages were treated with BafA1 or DMSO after infection with *S. aureus* WT, Δ*pvl* and complemented strain (compΔ*pvl*) for 4 h post infection. Total cell lysates were probed with antibodies against LukS-PV, a-hemolysin (Hla, control for *S. aureus* infections), and tubulin (loading control for macrophages). **(C–E)** Growth curve of *S. aureus* **(C)** WT, **(D)** Δ*pvl*, and **(E)** compΔ*pvl* in broth media with BafA1 or DMSO over

18 h. Untreated (UT) media without bacteria are controls Mean from three independent samples. The data underlying this Figure can be found in S1 Data.
(PDF)

**S1 Video. WT THP-1 macrophages were stained with LysoTracker Red and Draq7 (blue) and imaged every 15 min.**
(AVI)

**S2 Video. WT THP-1 macrophages were stained with LysoTracker Red and Draq7 (blue) and then treated with PVL toxin.** Cells were imaged every 15 min.
(AVI)

**S3 Video. SGMS1-deficient THP-1 macrophages (clone 1) were stained with LysoTracker Red and Draq7 (blue) and then treated with PVL toxin** . Cells were imaged every 15 min.
(AVI)

**S4 Video. SGMS1-deficient THP-1 macrophages (clone 2) were stained with LysoTracker Red and Draq7 (blue) and then treated with PVL toxin** . Cells were imaged every 15 min.
(AVI)

**S5 Video. WT THP-1 macrophages were stained with LysoTracker Red and Draq7 (blue) and then infected with GFP-expressing WT S. aureus (MOI = 10)** . Cells were imaged every 15 min.
(AVI)

**S6 Video. WT THP-1 macrophages were stained with TMRM (red) and Draq7 (blue) and then infected with WT *S. aureus*** . Cells were imaged every 15 min.
(AVI)

**S7 Video. WT THP-1 macrophages were stained with TMRM (red) and Draq7 (blue) and then infected with Δpvl mutant** . Cells were imaged every 15 min.
(AVI)

**S8 Video. WT THP-1 macrophages were stained with TMRM (red) and Draq7 (blue) and then infected with complemented Δpvl mutant** . Cells were imaged every 15 min.
(AVI)

**S9 Video. WT THP-1 macrophages were treated with Bafilomycin A1, stained with Draq7 (blue) and then infected with RFP-expressing WT *S. aureus*** . Cells were imaged every 15 min.
(AVI)

**S10 Video. WT THP-1 macrophages were treated DMSO, stained with Draq7 (blue) and then infected with RFP-expressing WT *S. aureus*** . Cells were imaged every 15 min.
(AVI)

**S1 Raw Images. Minimally unprocessed and uncropped gel images of Figs 1A, S1C, S9B, S13B and S14C.**
(PDF)

**S1 Table.**
(XLSX)

**S2 Table. The theoretical neutron scattering length density (SLD) of lipids and PVL.**
(DOCX)

**S3 Table. Fitted thickness and the calculated volume fractions of POPC/TOCL bilayer in the NR study with the binding of PVL in (A) pH 5.0 and (B) pH 7.4.**
(DOCX)

**S4 Table. Fitted thickness and the calculated volume fractions of POPC/POPA bilayer in the NR study with the binding of PVL in (A) pH 5.0 and (B) pH 7.4.**
(DOCX)

**S5 Table.**
(XLSX)

**S6 Table.**
(XLSX)

**S1 Data. All numerical values underlying the Figs 1 to S12.**
(XLSX)

**S2 Data. Gating strategy for Figs 2D, 3A and S9A.**
(PDF)

## Acknowledgments

We acknowledge the expert help from members of Monash FlowCore and Micro Imaging facilities. Purified recombinant LukAB was provided by Drs Cara Nethercott and Anton Peleg (Monash University) and the *S. aureus* wild-type strains SF8300 (USA300), isogenic Δpvl and complemented Δpvl strain by Dr Binh Diep (University of California, USA). The plasmid pHC48 was provided by Drs Jennifer Payne (Monash University), and Alexander Horswill (University of Colorado, USA). Anti-LAMP1 (H4A3) was deposited to the Developmental Studies Hybridoma Bank (DSHB) by August, J.T. and Hildreth, J.E.K. We thank the Australian Nuclear Science and Technology Organization (ANSTO) for the award of the neutron beamtime (proposal numbers P5342 and P6083).

## Author contributions

**Conceptualization:** Seong H. Chow, Thomas Naderer.

**Data curation:** Seong H. Chow, Yusun Jeon, Pankaj Deo, Amy T. Y. Yeung, Christine Hale, Sushmita Sridhar, Gilu Abraham, Joshua Nickson, Françios A. B. Olivier, Jhih-Hang Jiang, Yue Ding, Mei-Ling Han, Anton P. Le Brun, Dovile Anderson, Darren Creek, Janette Tong, Kip Gabriel, Ana Traven, Gordon Dougan, Hsin-Hui Shen, Thomas Naderer.

**Funding acquisition:** Hsin-Hui Shen, Thomas Naderer.

**Investigation:** Seong H. Chow, Joshua Nickson, Thomas Naderer.

**Methodology:** Hsin-Hui Shen.

**Project administration:** Thomas Naderer.

**Supervision:** Seong H. Chow, Jian Li, Hsin-Hui Shen, Thomas Naderer.

**Writing – original draft:** Seong H. Chow, Thomas Naderer.

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
