## [Editor Report · Decision Letter 0]

14 Jun 2024

Dear Dr Naderer, 

Thank you for submitting your manuscript entitled "Membrane rupture by the staphylococcal toxin Panton-Valentine leukocidin depends on phosphatidic acid, cardiolipin and acidic environments" for consideration as a Research Article by PLOS Biology.

Your manuscript has now been evaluated by the PLOS Biology editorial staff, as well as by an academic editor with relevant expertise, and I am writing to let you know that we would like to send your submission out for external peer review.

Once your full submission is complete, your paper will undergo a series of checks in preparation for peer review. After your manuscript has passed the checks it will be sent out for review. To provide the metadata for your submission, please Login to Editorial Manager (https://www.editorialmanager.com/pbiology) within two working days, i.e. by Jun 16 2024 11:59PM.

Kind regards,

Melissa

Melissa Vazquez Hernandez, Ph.D.

Associate Editor

PLOS Biology

---

## [Decision Letter · Decision Letter 1]

17 Jul 2024

Dear Dr Naderer,

Thank you for your patience while your manuscript "Membrane rupture by the staphylococcal toxin Panton-Valentine leukocidin depends on phosphatidic acid, cardiolipin and acidic environments" was peer-reviewed at PLOS Biology. It has now been evaluated by the PLOS Biology editors, an Academic Editor with relevant expertise, and by three independent reviewers, one of them Jos AG van Strijp. 

In light of the reviews, which you will find at the end of this email, we would like to invite you to revise the work to thoroughly address the reviewers' reports. As you can see below, as in the previous revision, the reviewers are concerned about the toxin concentrations used. After discussion with the Academic Editor and the reviewers, we do not require new experiments with a different concentration. However, the revised discussion should address and highlight how this concentration may not be relevant to in vivo infection. Additionally, the experimental requests and suggestions by reviewers #2 and #3 must be fulfilled for further consideration.

Given the extent of revision needed, we cannot make a decision about publication until we have seen the revised manuscript and your response to the reviewers' comments. Your revised manuscript is likely to be sent for further evaluation by all or a subset of the reviewers.

**IMPORTANT - SUBMITTING YOUR REVISION**

*Re-submission Checklist*

*Published Peer Review*

*PLOS Data Policy*

*Blot and Gel Data Policy*

Sincerely,

Melissa

Melissa Vazquez Hernandez, Ph.D.

Associate Editor

PLOS Biology

REVIEWERS' COMMENTS:

Reviewer #1: 

I have read the paper, the comments of the referees of the previous version and the response of the authors on their suggestions. In all, authors have addressed many different issues but for me the major issues have not been dealt with yet.

My major concern remains the concentration used. The authors claim that their high concentration can be observed in vivo, but that is not the issue. The issue is what is responsible for killing at the site of infection there is a halo of toxin surrounding viable bacteria, and cells (neutrophils and macrophages with the appropriate set of receptors) approach these bacteria and will be promptly killed at low concentrations (ng range), way before they reach the concentration of toxin used in these studies with liposomes (low to high ug range). So in my mind, the phenomena studied in this paper are not of importance for neutrophils and macrophages. They could be of importance in non-phagocytic cell (lacking the C5aR and CD45). The main question is how important that is in the pathophysiology of S. aureus infection. Also, because of the same reasons the title of the paper is highly confusing.

Although some of the experiments are impressive and well performed, the issue as illustrated above is not with the execution of the experiments but with the questions the authors pose and with the conclusions they draw. Overall the answers to the referees comments (i agree with all of the three referees) are not convincing to me. 

Reviewer #2: 

In their manuscript, Chow and colleagues nicely demonstrated the lipid binding partners of PVL in membranes, the requirement for sphingomyelin and cholesterol for internalization, and low pH for permeabilization of lysosome. They then performed S. aureus infection experiments with WT and PVL mutant, and bafilomycin to verify the role of PLV and low pH in S. aureus infectious physiology. This aspect of the study, even with the added data, still require more rigor.

Major comments

1) Sphingomyelin breakdown product ceramide is a second messenger in apoptotic cascade activation. Hence, KO of sphingomyelin could reduce cell death independently of its role in trafficking. Interpretation of findings linked to SGMS1 KO need to take that into account.

2) Although high concentration of PVL is measured in human samples, findings made on artificial membranes using higher PVL doses may not be relevant to what actually happens in the THP or standard macrophage assays, and this should be acknowledged by the authors.

3) Figure 6B and E. The authors propose that PVL should rupture the lysosome at lower pH and drive egress of S. aureus into the cytosol. Once in the cytosol, S. aureus should overgrowth (based on the video). Yet they state that there is no difference in intracellular CFU between WT and delta PVL (C), or control or bafilomycin (E). This does not fit the data.

Visually, it does look as if there is a difference between WT and delta PVL (C) and control versus bafilomycin (E) even is there is not noted statistical significance. Lack of significance could be attributed to lack of power. However, if that is the case, increased extracellular CFUs could simply be a reflection of increased intracellular CFUs. Irrespective, the experiment setup is not adequate to discriminate and verify the proposed hypothesis.

To fully test the hypothesis, the Bafilomycin experiment should really be performed using both WT and PVL KO to investigate if acidification related the findings (CFU and macrophage cell death) are directly related to PVL. In addition to measuring CFUs and (macrophage) cell deaths, the authors should quantify permeabilization of lysosome and mitrochondria in this experiment. 

4) The title of the paper implies that PVL lysis of macrophages depends on PA, cardiolipin and acidic environment. Demonstration of these PVL functions in the context of a S. aureus infection is currently not robust.

Minor comments

1) Concentration of purified toxins and S. aureus to host cell MOI used in assays should be noted in the figure legends.

4) The level of LukS-PV (MFI) bound to SGMS1 KO is half that bound to WT macrophages. While that likely does not substantially impact the conclusion drawn on the role of sphingomyelin in PVL trafficking, it is unclear if that would affect conclusion drawn on cell death if sphingomyelin is involved in cell death. This should be discussed. 

4) I agree that in absence of KO in host, it would be difficult to confirm the role of specific lipids in PVL interaction in SA infection. This has been noted as a limitation in the discussion.

Reviewer #3:

I consider that Chow et al. have addressed the main concerns raised by the reviewers during the initial consideration. Consequently, I find that the manuscript now fits well within the scope of the current forum.

However, I would like to point out one critical experiment that was not previously raised by the reviewers. It is standard practice in the field, to evaluate endolysosomal rupture using well-characterized markers such as Galectins (e.g., Galectin-3, Galectin-8) and the recruitment of endomembrane damage repair machinery (e.g., ESCRT, and now more recently PI4K2A, stress granules, etc.). Unfortunately, the current manuscript predominantly relies on Lysotracker images, which lack high resolution. Additionally, Lysotracker staining can be influenced by factors other than lysosomal membrane damage, such as pH neutralization or changes, which may occur independently of membrane damage. 

To strengthen the main findings, I recommend that the authors incorporate some of these standard studies, even if performed on fixed cells. For instance, studies have explored the link between endomembrane damage and mitochondrial function (10.1038/s41467-022-34632-8). Other relevant examples include: 10.1242/jcs.252973, 10.1016/j.devcel.2019.10.025, 10.7554/eLife.85727, 10.1038/s41586-023-06726-w, and 10.1371/journal.ppat.1007501. Incorporating such studies would significantly enhance the robustness and reliability of their conclusions. This would significantly enhance the robustness and reliability of their conclusions.

---

## [Decision Letter · Decision Letter 2]

4 Dec 2024

Dear Dr Naderer,

Thank you for your patience while we considered your revised manuscript "Recombinant staphylococcal Panton-Valentine leukocidin ruptures model membranes depending on phosphatidic acid, cardiolipin and acidic environments." for consideration as a Research Article at PLOS Biology. Your revised study has now been evaluated by the PLOS Biology editors, the Academic Editor and one of the original reviewers. 

In this case, both the Academic Editor and Reviewer #3 have evaluated the revised manuscript and agreed that most concerns have been addressed. However, they have raised a significant issue regarding Fig. S12 and the associated experiments, as outlined in Reviewer #3's report. To proceed with publication, we kindly request that you address the lysosome damage conern—potentially by repeating the experiment—, provide the additional controls and enhance the quality of the images in accordance with the reviewer’s suggestions.

**IMPORTANT - SUBMITTING YOUR REVISION**

*Resubmission Checklist*

*Published Peer Review*

*PLOS Data Policy*

*Blot and Gel Data Policy*

Sincerely,

Melissa

Melissa Vazquez Hernandez, Ph.D.

Associate Editor

PLOS Biology

REVIEWER'S COMMENTS:

Reviewer #3: 

Panel S12B now shows lysosome damage analysis using a GAL-3 puncta readout. One concern I have with the experiment is that LLOMe is a very well known cathepsin C substrate and, therefore, proteolytically activity-dependent. The use of an inhibitor of V-ATPase, such as bafilomycin, which will neutralize the lysosomes, should block its activity. However, the authors show no effect. Given the study's importance, the image resolution should be improved, ideally by showing multiple cells with a zoomed-in section to highlight the currently blurry puncta. The authors could enhance resolution by adjusting the acquisition parameters on their confocal system.

Minor:

Including additional markers such as LAMP-1 and DAPI would improve the panel. Figures do not present a scale bar.

The authors show lysosome damage and mitochondrial activity impairment phenotypes. This paper should be cited or discussed: 10.1038/s41467-022-34632-8.

---

## [Decision Letter · Decision Letter 3]

28 Jan 2025

Dear Dr Naderer,

Thank you for your patience while we considered your revised manuscript "Recombinant staphylococcal Panton-Valentine leukocidin ruptures model membranes depending on phosphatidic acid, cardiolipin and acidic environments." for publication as a Research Article at PLOS Biology. This revised version of your manuscript has been evaluated by the PLOS Biology editors, the Academic Editor [and the original reviewers -EDIT AS APPLICABLE].

Based on the reviews, we are likely to accept this manuscript for publication, provided you satisfactorily address the remaining editorial points. Please also make sure to address the following data and other policy-related requests.

a) We routinely suggest changes to titles to ensure maximum accessibility for a broad, non-specialist readership, and to ensure they reflect the contents of the paper. In this case, we would suggest a minor edit to the title, as follows. Please ensure you change both the manuscript file and the online submission system, as they need to match for final acceptance:

"Staphylococcal toxin PVL ruptures model membranes under acidic conditions through interactions with cardiolipin and phosphatidic acid"

b) There seem to be an error in the abstract. Specifically the following: "rather an plasma membrane binding" - this does not make sense, perhaps there is a part missing?

Please supply the numerical values either in the a supplementary file or as a permanent DOI’d deposition for the following figures:

Figure 1B-H, 2A-G, 3FG, 4B-E, 5, 6ABCEFG, S1AB, S2ABC, S3A-H, S4A-F, S5A-D, S6A-J, S7BCD, S8A-H, S12B

d) Please cite the location of the data clearly in all relevant main and supplementary Figure legends, e.g. “The data underlying this Figure can be found in S1 Data” or “The data underlying this Figure can be found in https://doi.org/10.5281/zenodo.XXXXX”

e) Thank you for providing some of the uncropped gels. However, we still require the original, uncropped and minimally adjusted images supporting all blot and gel results reported in the Figures1A, S1C, S9B. I would also like to point out that is is not clear on the second page of the S1_raw_images file, to which Figure it corresponds. Additionally, it may be the case that S14C is mislabeled.

f) For figures containing FACS data (Figures 3AB and S9A), please provide the FCS files and a picture showing the successive plots and gates that were applied to the FCS files to generate the figure. We ask that you please deposit this data in the FlowRepository (https://flowrepository.org/) and provide the accession number/URL of the deposition in the Data Availability Statement in the online submission form.

g) Please ensure that your Data Statement in the submission system accurately describes where your data can be found and is in final format, as it will be published as written there.

h) Per journal policy, if you have generated any custom code during the course of this investigation, please make it available without restrictions upon publication. Please ensure that the code is sufficiently well documented and reusable, and that your Data Statement in the Editorial Manager submission system accurately describes where your code can be found.

We expect to receive your revised manuscript within two weeks. 

*Published Peer Review History*

*Press*

Sincerely,

Melissa

Melissa Vazquez Hernandez, Ph.D.

Associate Editor

PLOS Biology

REVIEWERS' COMMENTS

Reviewer #3: 

Although I partially disagree with many of the comments regarding the mechanism of action of LLOMe and interpretations based on previous reports, I consider the authors have now addressed the remaining issues and am supportive of this publication.

To note:

*The effect of cathepsin C on LLOMe is clearly demonstrated in the Thiele/Lipsky PNAS paper (10.1073/pnas.87.1.83). The manuscript the authors mentioned prior to this work (10.1016/0005-2736(73)90114-4) has no experiments conducted in cellulo and it does not use LLOMe (L-leucyl-L-leucine methyl ester) but L-leucine methyl ester. Also, Thiele/Lipsky do not compare NH4Cl (which is a common reagent in many cell lysis buffers and has broader, less specific effects than those achieved by targeting lysosomes) with BAFA1.

*The dynamics of lysosome membrane damage and recognition by galectins are highly heterogeneous, as are lysosomes, and therefore generalizations should be carefully considered. Please see 10.1371/journal.pbio.3002576 and 10.1083/jcb.202403116.

*In a similar line, the lysosome protease content highly varies among tumor-derived cell lines (such as the one used in the JCS paper mentioned by the authors) and specialized cells such as macrophages (10.1126/science.1108003). Therefore, the kinetics of membrane damage recognition and LLOMe proteolytic processing should not be expected to be comparable.

*In the Florey et al. paper indicated by the authors, where bafilomycin is used as a pre-treatment and still gal-3 puncta are observed, we should note that gal-3 puncta are not quantified in that figure, and we do not have data of any titration curve, etc. Given there are no gal-3 puncta quantifications, we can't conclude if there is a statistically significant reduction or not. Please note this is the only manuscript of all the ones mentioned before that uses BAFA1 as a pre- and/or co-treatment. So, I would carefully consider over-interpretations.

*As the authors suggested, the cell type, reagent treatment (pre-treatment only vs. pre- and co-treatment, etc.), and reagent concentration should be analyzed. For instance, in the Maejima et al. paper presented here by the authors, bafilomycin is only added after LLOMe treatment and not as a pre-treatment because the focus is on the autophagy machinery and not on the lysosomal proteases or LLOMe processing.

*Collectively, this also highlights limitations of generalized conclusions solely based on the use of chemical inhibitors. For example, in the Bussi et al. paper, the conclusions are supported by using specific lysosome protease KO macrophages and other cell KOs.

---

## [Editor Report · Decision Letter 4]

24 Feb 2025

Dear Thomas,

Thank you for the submission of your revised Research Article "Staphylococcal toxin PVL ruptures model membranes under acidic conditions through interactions with cardiolipin and phosphatidic acid" for publication in PLOS Biology. On behalf of my colleagues and the Academic Editor, Maximiliano Gutierrez, I am pleased to say that we can in principle accept your manuscript for publication, provided you address any remaining formatting and reporting issues. These will be detailed in an email you should receive within 2-3 business days from our colleagues in the journal operations team; no action is required from you until then. Please note that we will not be able to formally accept your manuscript and schedule it for publication until you have completed any requested changes.

PRESS

Sincerely, 

Melissa

Melissa Vazquez Hernandez, Ph.D., Ph.D.

Associate Editor

PLOS Biology
